JCB | Journal of Cell Biology

# Perinuclear non-centrosomal microtubules direct nuclei dispersion during epithelial morphogenesis

Rashmi Budhathoki[1], Liam J. Russell[1,2], Dinah Loerke[2], and J. Todd Blankenship[1]

**As cells contract and reshape to enable tissue morphogenesis, their own internal structures can constrain these behaviors. In the *Drosophila* germband, the uncrowding of nuclei away from an initially common plane is required for efficient cell intercalation and extension. Here, we find that a centrosomally derived microtubule network transitions into non-centrosomal arrays that are deeply embedded in nuclei before shifting towards the apical cortex as GBE progresses. Disrupting ncMT function by compromising *CLASP* or *Patronin* function leads to failures in nuclear dispersion and results in MT networks dominated by centrosomal arrays. *CLASP* disruption also causes a marked detachment of MTs from nuclei, severely affecting nuclear orientation and dispersion. Our results also reveal a fundamental antagonism between ncMT and centrosomal networks—an observation corroborated in *γ-tubulin* embryos. Lastly, *EB1* disruption blocks the apical shift of ncMTs, leading to dispersion defects. Overall, our findings reveal that nuclear repositioning during epithelial remodeling depends on a centrosome-to-ncMT transition requiring *CLASP*, *EB1*, and *Patronin* function.**

## Introduction

Building the tissue architectures necessary for complex organismal function calls for cells to adopt a variety of shapes during development. The nucleus, being the largest and most mechanically rigid organelle in the cell, permits epithelial cell shape changes through either deformation or repositioning pathways (Lammerding, 2011; Tocco et al., 2018; Janota et al., 2020; de Leeuw et al., 2024). Additionally, nuclei act as mechanosensors and respond to cell shape changes by altering gene expression or modulating actomyosin dynamics and cellular plasticity (Maurer and Lammerding, 2019; Venturini et al., 2020). Failure to adapt to changes in cell shape increases nuclear strains, which may lead to nuclear blebbing and rupture and the eventual damage to genetic material (Srivastava et al., 2021). There is therefore a growing appreciation for the need of cells to coordinate intracellular organization with cell shape dynamics during tissue development and homeostasis.

This relationship between cell and nuclear shapes is very apparent during early morphogenesis, where rapid large-scale tissue rearrangements are driven by coordinated cell shape changes. In the *Drosophila* gastrula, epithelial germband cells undergo dramatic shape changes that direct polarized cell intercalation and extension of the tissue along the anterior-posterior axis (Irvine and Wieschaus, 1994; Bertet et al., 2004; Blankenship et al., 2006; Rauzi et al., 2010; Jewett et al., 2017). These mechanical changes to the plasma membrane require not

only the remodeling of the underlying actin cortex but also the reorganization of the cell's internal structures. Nuclei, which can act as mechanical bumpers, need to be accommodated by cells in such a way that they do not oppose the cell shape changes required for intercalation (de Leeuw et al., 2024). Prior to the onset of tissue extension, cells in the epithelium are largely hexagonal and nuclei are positioned in a common apical plane in the columnar epithelia. As germband extension (GBE) begins the rapid transformation in cell topologies that drive intercalation, nuclei also adapt to the new cell shapes through shape deformations and by repositioning into alternate, and deeper, apical–basal planes (de Leeuw et al., 2024).

Microtubules (MTs) are one of the key cytoskeletal structures that have been implicated in controlling nuclear architecture, their placement within cells, and genome organization (Tran et al., 2001; Zhao et al., 2012; Tremblay et al., 2013; D'Alessandro et al., 2015; Ramdas and Shivashankar, 2015; Geng et al., 2023). MT networks can be broadly categorized into centrosomal and non-centrosomal (nc) networks depending on their origin. Centrosomes are MT-organizing centers at which γ-tubulin–mediated nucleation of MTs occurs through the assembly of α- and β-tubulin heterodimers. However, MT networks also operate in cells lacking functional centrosomes (Megraw et al., 2001; Dumont and Desai, 2012; Wolff et al., 2016; Chinen et al., 2020). These ncMT networks may arise from apical plasma membranes, Golgi

---

[1]Department of Biological Sciences, University of Denver, Denver, CO, USA;  [2]Department of Physics and Astronomy, University of Denver, Denver, CO, USA.

Correspondence to J. Todd Blankenship: todd.blankenship@du.edu.



structures, nuclear envelopes, chromosomes, and pre-existing MTs (Dumont and Desai, 2012; Wu and Akhmanova, 2017). In other cases, MTs nucleated from centrosomes may be severed, released, and then stabilized as non-centrosomal populations by the activity of Patronin/CAMSAP family proteins (Jiang et al., 2014; Sanchez and Feldman, 2017; Gillard et al., 2021).

In the early *Drosophila* syncytium, centrosomal MT networks form an inverted basket tightly wrapping interphase nuclei and are necessary for nuclear apicobasal elongation and deformation during *Drosophila* cellularization (Warn and Warn, 1986; Schejter and Wieschaus, 1993; Harris and Peifer, 2007; Hampoelz et al., 2011). These centrosomal networks also rearrange into spindle arrays to direct the cleavage divisions, and dysfunctional centrosomes result in aberrant spindles, causing chromosomal missegregation and mitotic failures (Wilson and Borisy, 1998; Megraw et al., 1999; Stevens et al., 2007; Alvarez-Rodrigo et al., 2019; Rollins and Blankenship, 2023). During the epithelialization of the embryo just prior to gastrulation and tissue extension, the MT bundles that form the basket begin to dissociate from the centrosomes (Harris and Peifer, 2007) to form parallel arrays in the apical–basal axis. Several recent studies have focused on apical MT function in activating and controlling actomyosin networks in the epithelium and during folding morphogenesis (Takeda et al., 2018; Garcia De Las Bayonas et al., 2019; Ko et al., 2019; Roby and Rauzi, 2025). However, how these MT networks mediate the redistribution of nuclei necessary for epithelial tissue extension is unclear.

In this study, we identify the rapid changes in MT networks that occur in the early epithelium and which regulate nuclear dispersion dynamics during GBE. Our results demonstrate that newly generated ncMTs are deeply embedded alongside GBE nuclei. As these perinuclear MTs shift apically and away from nuclei, they become cortically anchored and nuclei move to deeper basal regions. Disrupting MT reorganization by compromising Patronin function causes a loss of perinuclear and apical MT networks, leading to nuclear dispersion defects. Perturbing MT +end binding protein function in *CLASP* embryos also led to the depletion of ncMTs and defects in the anchoring, orientation, and dispersion of nuclei. Intriguingly, our data also reveal an antagonistic relationship between centrosomal and ncMT networks. Lastly, perturbing EB1 +TIP function prevented the apical shift of the ncMTs and nuclear dispersion, implicating EB1 in the cortical anchoring of the perinuclear ncMT array. Altogether, these data demonstrate that the rapid transition of MT networks from centrosomal to an apically anchored ncMT organization directs the dispersion of nuclei necessary for tissue remodeling.

## Results

### MT networks undergo continuous remodeling during GBE
At the onset of tissue extension in the *Drosophila* embryo, nuclei undergo a rapid redistribution along the apical–basal axis to facilitate efficient cell intercalation (Fig. S1 A) (de Leeuw et al., 2024). As the tissue remodels, nuclei initially positioned in common apical planes move into deeper basal cell regions, causing an approximate halving of nuclear densities in apical

regions (Fig. S1, A and A′). Our previous work has shown that MTs are essential in this nuclear dispersion process (de Leeuw et al., 2024), but how MTs mediate this movement is not known. Additionally, which of the several populations of MT networks present in the embryonic epithelium direct this movement is unclear. As a first step in this analysis, we imaged MTs (Jupiter: GFP) with an RFP-NLS to visualize nuclei. As reported previously (Warn and Warn, 1986; Schejter and Wieschaus, 1993; Harris and Peifer, 2007; Hampoelz et al., 2011), prior to the onset of cell intercalation, MTs are largely centrosomal in origin and form inverted MT baskets that are closely associated with nuclei (Fig. 1, A–A‴). When GBE initiates, MT bundles are still closely associated with nuclei (Fig. 1 B′). However, many of these perinuclear MTs appear to be separated from the centrosome, suggesting the presence of a growing population of ncMTs (Fig. S1 B). Importantly, as cell intercalation proceeds, we observed three major MT remodeling events: (1) MT enrichment at centrosomes was depleted by ~35% and MT bundles completely detach from centrosomes (Fig. 1, A–D; Fig. S1 C; and Video 1); (2) intensities of the prominent perinuclear MT bundles that wrap nuclei weaken (64% decrease as measured at nuclear midplanes; Fig. 1, C′–C‴ and E), and (3) MT bundles populate the apical non-nuclear regions either by apical sliding of perinuclear bundles or by disassembly of perinuclear MT pools followed by recruitment at apical regions (Fig. 1, A″–C″ and F). Thus, nuclei lose their basket of closely associated MTs during the dispersion of nuclei. Additionally, these data indicate that MT networks move upwards and away from nuclei (and centrosomes) and into nuclei-free apical regions, specifically as tissue extension and nuclear dispersion advances.

Given the differential persistence of MT pools at centrosomes and perinuclear regions, we next wanted to test if these behaviors potentially denote networks of differing MT stabilities. To this end, we performed immunostaining with anti-acetylated α-tubulin antibodies that are indicative of stable MT populations. We observed that the perinuclear MT population was brightly labeled with the antibody, which is absent from the centrosomes or the apical cell cortex (Fig. 1 G; and Fig. S1, D and D′). These data are further supported by fluorescence recovery after photobleaching (FRAP) experiments during early GBE, where we observed rapid MT recovery (α-tubulin:GFP) at centrosomes with a mean half-life of 5.1 s. By contrast, MTs in perinuclear regions possessed a mean half-life of 18.2 s (Fig. 1, H–H″). Nearly 40% of the perinuclear MT population failed to recover, while only a small population of ~15% did not recover at the centrosomes (Fig. 1 H‴). Finally, colchicine-injected embryos displayed a strong depletion of centrosomal and non-nuclear MT, but perinuclear MTs persisted, revealing a resistance to colchicine treatments (Fig. S1 E). These results suggest that ncMTs form a hyperstable network that is found in close association with nuclei, while centrosomally derived MTs are much more dynamic and appear to be under a state of constant flux.

### Perinuclear and apical pools of MTs are disrupted in Patronin-compromised embryos
Given the potential correlation of ncMT behaviors and nuclear dynamics, we examined the function of Patronin, a CAMSAP and

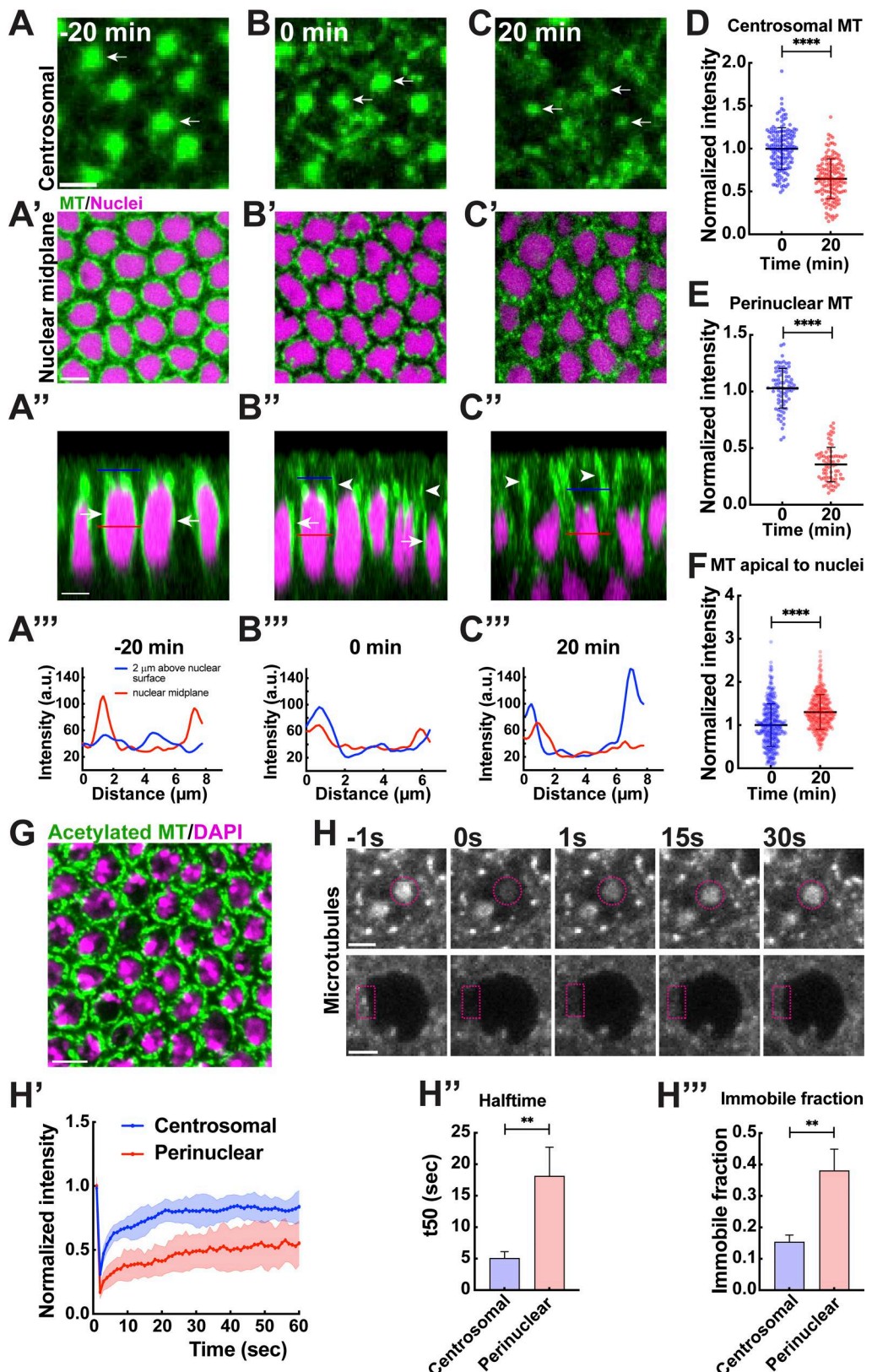

Figure 1. **MTs undergo a centrosomal to non-centrosomal transition during GBE. (A–C)** MTs marked with Jupiter:GFP at level of centrosomes 20 min pre-GBE (A), at GBE onset (B), and 20 min into GBE (C). Arrows point to centrosomal MT. **(A'–C')** Perinuclear MT pools (green) at 20 min pre-GBE (A'), at onset (B'), and 20 min into GBE (C'). **(A''–C'')** Orthogonal projection of MT and nuclei in epithelium 20 min before GBE (A''), at onset (B''), and 20 min into GBE (C''). **(A'''–C''')** Line scan intensity plot for MT as denoted in the lines (blue = 2 μm above nuclear midplane, red = at nuclear midplane) shown in A'', B'', and C'', respectively. **(D)** Normalized intensity of centrosomal MTs at 0 and 20 min GBE; n = 150 centrosomes for each time point, k = 3 embryos. **(E)** Normalized intensity

of perinuclear MTs at 0 and 20 min GBE; *n* = 80 and 73 perinuclear regions for 0 and 20 min, respectively, *k* = 3 embryos. **(F)** Normalized intensity of apical MTs at 0 and 20 min; *n* = 396 and 377 regions for 0 and 20 min, respectively, *k* = 3 embryos. **(G)** Stable perinuclear MT pools indicated by brightly stained acetylated MT (green) around nuclei (DAPI, magenta). **(H)** Still frames showing FRAP of α-tubulin:GFP at centrosomes (top) and perinuclear region (bottom). Time is indicated in seconds; photobleaching was performed at 0 s. Circle or rectangle in magenta indicates the photobleached region. **(H′)** Fluorescence recovery profile for centrosome and perinuclear regions. **(H′′)** Halftime of recovery for centrosomal vs perinuclear α-tubulin:GFP (mean ± SEM). **(H′′′)** Immobile fraction for centrosomal vs perinuclear α-tubulin:GFP (mean ± SEM). For (H′–H′′′), *n* = 11 FRAP regions for each plot, *k* = 11 embryos. Scale bar = 5 µm for (A′) and (G), 3 µm for (A) and (A′′), and 2 µm for (H). All scatter plots show the mean ± SD. Statistical significance was calculated using the Mann–Whitney U-test. **P < 0.01 and ****P < 0.0001.

MT minus-end binding (−TIP) protein, often implicated in stabilizing ncMT populations. Consistent with the existence of a nucleus-associated ncMT population, we observed a striking depletion of perinuclear MT pools in *Patronin*-disrupted (shRNA) embryos as compared to control measurements (Fig. 2, A and B). Additionally, instead of tightly wrapping the nuclei at the GBE onset, the remaining perinuclear MT bundles were detached from the nuclei after Patronin disruption (Fig. 2 A, arrows). These embryos also displayed a highly diminished apical MT population (decreased by 50.7%) as compared to controls (Fig. 2 C, arrows; Fig. S2 A). Interestingly, not all MT populations decreased in *Patronin* embryos—centrosome-associated MTs were *enhanced* by ∼32% at GBE onset and by ∼40% at 20 min into GBE as compared to control measurements (Fig. 2, D and E; and Video 2). We next examined if the semi-detached perinuclear MTs observed after *Patronin* disruption showed stabilities that were more characteristic of centrosomal MTs. Indeed, perinuclear MT pools in *Patronin* embryos recovered three times faster than perinuclear MTs in control embryos, and the *Patronin* immobile fraction decreased to 21% from 38% in control embryos (Fig. S2, B–E). *Patronin* centrosomal MT recovery rates or immobile fractions were not significantly different from control measurements, suggesting that *Patronin* has little function in regulating this population of MTs (Fig. S2 D). We also observed that acetylated MT levels after *Patronin* disruption were decreased and appeared fragmentary in nature (Fig. S2, F and F′). At the protein level, Patronin:GFP also undergoes a redistribution from an initially perinuclear localization to a cortical location as tissue extension proceeds, similar to the observed shifts in ncMT populations (Fig. 2, G and G′) (Ko et al., 2019; Takeda et al., 2018). Patronin:GFP is additionally found near centrosomes and this population depletes during GBE (Fig. 2 H). These observations demonstrate that perinuclear MTs are stabilized by Patronin and likely constitute a ncMT population that shifts away from nuclei to an apical location as GBE occurs.

## Patronin disruption inhibits active nuclear dispersion

We next wanted to determine the impact of the ncMT function on nuclear positioning and dispersion in the germband epithelium. Examining nuclear behaviors after *Patronin* disruption suggests that ncMTs play an essential function in the dispersion of nuclei from apical regions into deeper basal portions of the cell. In control embryos, approximately half (44%) of nuclei move into cell regions deeper than 10 µm from the apical surface within 20 min of the start of cell intercalation (Fig. 2, F and G). However, in *Patronin* embryos, 91% of the nuclei remain in the top apical 10 µm of the cell, indicative of a highly defective nuclear dispersion process in these embryos (Fig. 2, F and G). Additionally, and consistent with the detachment of perinuclear

MTs observed in *Patronin* embryos, nuclei appeared to have lost their ability to position themselves in this apical region (Fig. 2 C). In control embryos, nuclei did not approach the apical cortex and maintained a distance of at least 2 µm from the apical surface throughout GBE. After *Patronin* disruption, 25% of nuclei entered into this apical exclusion zone as compared to 2% in control embryos (Fig. 2, C and H). The speed at which nuclei move was also grossly disrupted in *Patronin* embryos—peak speeds of 1.31 µm/min are observed in control nuclei, but this is reduced to 0.66 µm/min in *Patronin* embryos (Fig. S2 I). Similarly, average speeds were reduced from 0.44 µm/min in controls to 0.16 µm/min in *Patronin* embryos (Fig. 2 I). We also used mean squared displacement (MSD) analysis to detect periods of active motion—this showed that *Patronin* nuclei spend less time actively dispersing (46% of nuclei had periods of active motion as compared to 71% in control embryos) (Fig. 2 J). Thus, the loss of ncMT function due to Patronin disruption affects the speed and frequency at which nuclei move within cells and severely obstructs nuclear dispersion.

## MT +end binding protein CLASP embryos show ncMT organization defects

We were next interested if +TIP mediated anchoring and stabilization of ncMTs is an essential part of orienting nuclei and directing nuclear dispersion during GBE. CLASP is a MT +end tracking protein (+TIP) that has been implicated in MT-cortex interaction and could be a candidate to mediate cortical associations of ncMTs for dispersion (Mimori-Kiyosue et al., 2005; Lansbergen and Akhmanova, 2006; Ambrose and Wasteneys, 2008; Ambrose et al., 2011). Indeed, compromising *CLASP* function showed a disruption of MT networks at the onset of the GBE (Fig. 3 A). Further, these defects resembled that of *Patronin* embryos in numerous ways. First, perinuclear MT pools were depleted and detached away from the nucleus at the onset of the GBE (Fig. 3, A and B). Our quantification showed depletion of perinuclear MTs by ∼60% at GBE onset (Fig. 3 B). Second, we observed that later onset apical MT pools were also diminished (Fig. 3 C, arrows; Fig. S3 A). Lastly, MTs were highly enriched at centrosomes as compared to control embryos, and these centrosomal MTs were present deeper into GBE than in controls (Fig. 3 C arrowheads, Fig. 3, D and E; Fig. S3 B; and Video 3). Centrosomal MT intensities were ∼53% higher in *CLASP* embryos than in control embryos at GBE onset, and ∼79% higher at 20 min into the GBE (Fig. 3 E). Compared to *Patronin* embryos, the centrosomal MT intensity was increased by ∼16% at the onset and ∼28% at 20 min into GBE in *CLASP* embryos (Fig. S3 C). CLASP:GFP was found at the apical plasma membrane, centrosomes, and around the nuclear periphery (Fig. S3 D). Embryos

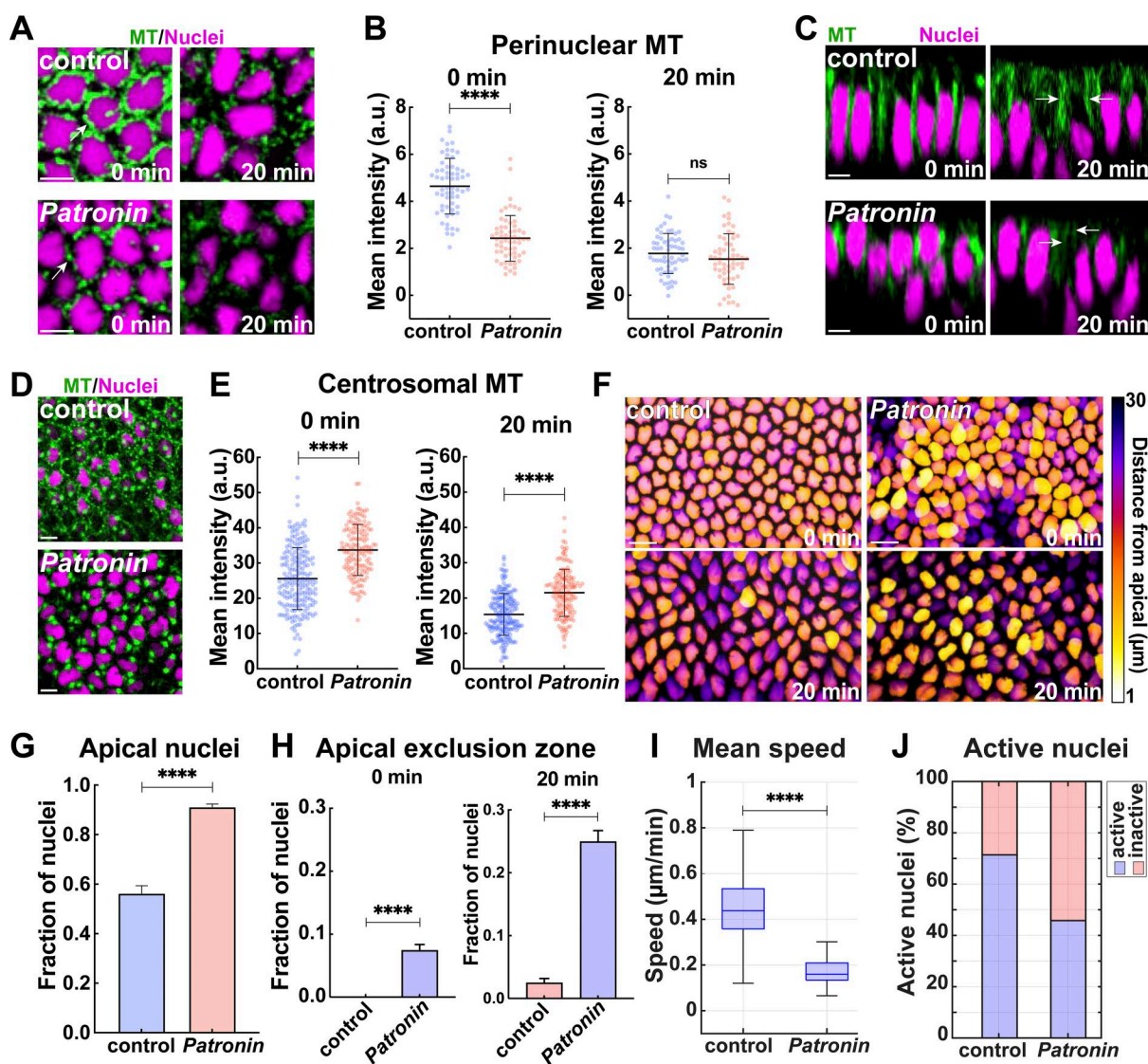

Figure 2. **Compromising Patronin function disrupts ncMT pools and severely impedes nuclear dispersion. (A)** Still frames showing MT depletion and detachment from nuclei in Patronin shRNA (*Patronin*) embryos compared to control. **(B)** Quantitation highlighting the depletion of perinuclear MT intensities in *Patronin* embryos compared to control at GBE onset; *n* = 60 perinuclear regions for each background at 0 min and *n* = 61 and 59 perinuclear regions for control and *Patronin* measurements, respectively, at 20 min, k = 3 embryos each. **(C)** Apical–basal view of MTs and nuclei in control (top) and *Patronin* (bottom) embryos at 0 and 20 min, showing decreased apical MT networks after Patronin disruption. Arrows point to apical MT. **(D)** Still frame showing MT enrichment at centrosomal regions in *Patronin* embryos as compared to control embryos. **(E)** MT intensities at centrosomes in control and *Patronin* embryos at 0 and 20 min, indicating enhanced centrosomal MT at both time points; *n* = 200 and 150 centrosomal regions for control and *Patronin*, respectively, at 0 min, and *n* = 200 and 149 centrosomal regions for control and *Patronin*, respectively, at 20 min, k = 3 embryos for each background. **(F)** Maximum-intensity projections of nuclei with color-codes based on distance from cell apices in control and *Patronin* embryos at onset and 20 min into GBE. **(G)** Comparison of the fraction of nuclei present in the apical 10 μm of cells in control and *Patronin* embryos indicating inhibited nuclear dispersion on disruption of ncMT; *n* = 263 and 605 nuclei from k = 3 embryos for control and *Patronin*, respectively. **(H)** Fraction of nuclei present in the apical-most 2 μm of the cell (apical exclusion zone) in control and *Patronin* embryos; *n* = 455 and 833 nuclei in control and *Patronin*, respectively at 0 min and *n* = 483 and 605 nuclei in control and *Patronin*, respectively, at 20 min, k = 3 embryos for each background. **(I)** Mean nuclear speed in control and *Patronin* embryos. **(J)** Percent of active nuclei as detected by the mean squared displacement (MSD) metric. **(I and J)** *n* = 546 and 291 measured nuclei for control and *Patronin*, respectively, k = 3 embryos each background. Scale bar = 10 μm for (F), 5 μm for (A, C, and D). Fig. 2, A, C, D, and F; Fig. 5 E; Fig. 3, C, D, and F; Fig. 5 H; and Fig. 7 C, control images/plots reproduced for comparison purposes. All scatter plots show the mean ± SD. Statistical significance was calculated using the Mann–Whitney U-test. ns, not significant. ****P < 0.0001.

co-expressing *CLASP* shRNA and α-tubulin:GFP could not be recovered, precluding FRAP analysis, but imaging revealed the loss and fragmentation of acetylated MTs, consistent with a destabilization of ncMTs after CLASP disruption (Fig. S3, E and E′-arrows). Further, MT bundles emanating from the centrosomes did not appear to be acetylated since immunostaining did not reveal such extended bundles. In total, MT organization in *CLASP* embryos resembled *Patronin* embryos, albeit with even higher centrosomal MT enrichments and with more fragmented acetylated perinuclear MT. These results suggest that CLASP +TIP binding aids in the stabilization and localization of ncMTs around and above the nucleus.

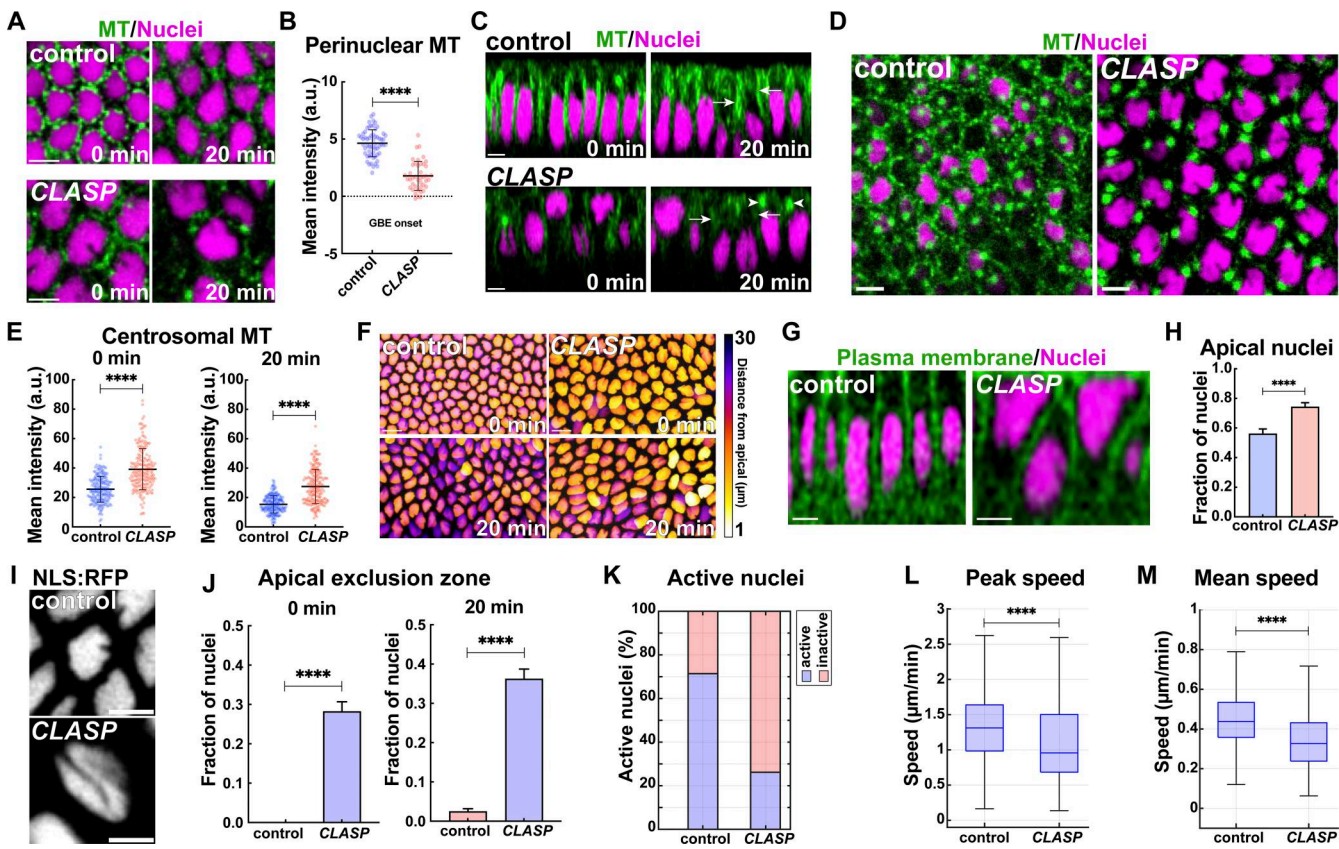

Figure 3.   **Disrupting CLASP function compromises formation of ncMT perinuclear baskets and nuclear orientation and dispersion. (A)** Still frames comparing perinuclear MTs in control and CLASP shRNA (*CLASP*) embryos reveal depleted MTs and detachment from nuclei. **(B)** Scatter plot of perinuclear MT intensities at GBE onset after *CLASP* disruption; n = 60 and 42 perinuclear regions for control and *CLASP*, respectively, k = 3 embryos each background. **(C)** Orthogonal projection showing depleted MTs (green) in *CLASP* embryos and defective nuclear positioning as compared to control. Arrows mark apical MTs and arrowheads mark centrosomal MTs, respectively. **(D)** Still frames showing enhanced centrosomal MTs and depleted perinuclear MTs in *CLASP* embryos compared to control. **(E)** Quantitation reveals enhanced MT intensities at centrosomes in *CLASP* as compared to control at 0 and 20 min GBE; n = 200 centrosomal regions for each background, k = 3 control embryos and k = 4 *CLASP* embryos. **(F)** Maximum-intensity projection of nuclei color-coded for position from cell apices in control and *CLASP* embryos. **(G)** Orthogonal projection of cells and nuclei in control and *CLASP* embryos showing apically collapsed and deformed nuclei in *CLASP* embryos. **(H)** Fraction of nuclei in apical 10 μm of cell showing nuclear crowding in apical regions in *CLASP* embryos; n = 236 and 258 for control and *CLASP*, respectively, from k = 3 embryos. **(I)** Still image of nucleus displaying misaligned groove after *CLASP* disruption. **(J)** Fraction of nuclei invading the apical exclusion zone in control and *CLASP* embryos; n = 455 and 340 nuclei for control and *CLASP*, respectively, at 0 min and n = 483 and 378 nuclei for control and *CLASP*, respectively, at 20 min, k = 3 embryos each. **(K)** Percent of active nuclei as detected by MSD in control and *CLASP* embryos. **(L)** Peak nuclear speeds in control and *CLASP* embryos. **(M)** Mean nuclear speeds in control and *CLASP* embryos. (J–L), n = 546 and 427 measured nuclei in control and *CLASP*, respectively, k = 3 embryos for each background. (M) Scale bar = 5 μm in A, C, D, G, I, and 10 μm in F. Fig. 3, A, C, and D; Fig. 7, A and C; Fig. 2, C and D; and Fig. 5 H reproduced for comparison purposes. All scatter plots show the mean ± SD. Statistical significance was calculated using the Mann–Whitney U-test. ****P < 0.0001.

## CLASP perturbation causes nucleus positioning and dispersion defects

Next, we wanted to examine how nuclear behaviors are altered in these embryos with disrupted ncMT networks. Unlike in control embryos, where nuclei disperse from a common apical plane at the onset of GBE, nuclei in *CLASP* embryos remain enriched in apical cell region—74% of *CLASP* nuclei remain in the apical-most 10 μm versus ~56% of nuclei in control embryos (Fig. 3, F–H). Additionally, nuclei in *CLASP* embryos appear to have difficulties in properly orienting and anchoring themselves in the apical cytoplasm (Fig. 3, G and I). *CLASP* nuclei often appeared to have been pushed against the apical cortex, causing a deformation of the nuclei (Fig. 3, F and G), and rotations of nuclei showing "hot dog" grooves that are normally oriented along the apical–basal axis were apparent (Fig. 3 I). These cortex-collapsed

nuclei also failed to maintain their basally elongated shapes and instead possessed rounder and more irregular shapes (Fig. 3, C and G). In line with these observations, we detected an increased percentage of *CLASP* nuclei invading the apical exclusion zone that worsened over time (~28% at the onset and ~36% at 20 min of GBE) (Fig. 3 J). Nuclei in *CLASP* embryos rarely possessed periods of active movements (25% of nuclei had MSD-based active motion compared to 71% of control nuclei) and peak and average speeds were decreased (0.96 μm/min peak speed in *CLASP* versus 1.31 μm/min peak speed in control; 0.33 μm/min average speed in *CLASP* versus 0.44 μm/min in control) (Fig. 3, K–M). These results are consistent with a ncMT population that is stabilized and anchored by +TIP proteins—the loss of this CLASP +TIP function leads to a failure to properly disperse and position nuclei (Fig. 3, F–H and J). In turn, this can lead to nuclei

being pressed against cell cortices causing a resultant disruption in nuclear shape.

## ncMT and centrosomal MT antagonism revealed in Patronin- and CLASP-disrupted embryos

As –TIP Patronin proteins and +TIP CLASP proteins function through distinct mechanisms, we wanted to more closely examine the changes in MT networks after disruption of each of these proteins. Particularly striking was the enhancement of centrosomal MTs, which could not rescue dispersive nuclear behaviors, and occurred in both backgrounds. We first examined the levels and localization of γ-tubulin, one of the key regulators of centrosomal PCM nucleation and a minus-end binding protein. Patronin-compromised embryos displayed elevated levels of γ-tubulin:GFP at centrosomes (1.6-fold at the GBE onset and 2.5-fold at 20 min GBE as compared to the control embryos) (Fig. 4, A and B), consistent with the persistent and elevated MT levels observed in these embryos. Interestingly, CLASP-disrupted embryos had a *depletion* of γ-tubulin intensity by ~28% at GBE onset (Fig. 4, A and B). These results indicate that Patronin depletion causes upregulation of centrosomal γ-tubulin, potentially suggesting an antagonism between ncMTs and centrosomal MT formation. However, the loss of γ-tubulin from centrosomes after CLASP disruption seemed at odds with the observed persistence and enhancement of centrosomal MTs in these embryos. As Patronin was a critical regulator of MT stabilities outside the centrosome, and given a potential antagonism between ncMT and centrosome function, we wondered if Patronin could be compensating for lower γ-tubulin levels in *CLASP* embryos. Examining Patronin:GFP in *CLASP*-disrupted embryos revealed a major shift in Patronin localization—perinuclear Patronin was lost even before the onset of GBE and further depleted at cortical pools where it is normally found later in GBE (Fig. 4, C and D). However, Patronin enrichment and persistence at centrosomes is much more robust than in the control embryos (Fig. 4, E and F; and Fig. S4, A and B). Centrosomal Patronin:GFP increased by 83% at GBE onset and by 48.5% at 20 min GBE after *CLASP* disruption as compared to control embryos (Fig. 4 F). These results implicate CLASP in stabilizing both ncMTs and γ-tubulin recruitment at the centrosome, as well as suggesting a potential antagonistic relationship between centrosomal and ncMT function.

## γ-tubulin disruption leads to ectopic centrosomal Patronin recruitment

Our above observations of Patronin upregulation at centrosomes after γ-tubulin depletion in *CLASP* embryos and of enhanced centrosomal γ-tubulin after Patronin depletion led us to directly examine embryos with γ-tubulin disruption (*γ-tubulin37C* shRNA). Intriguingly, Patronin:GFP in γ-tubulin perturbed embryos revealed a dramatic increase in overall Patronin intensities (Fig. 5 A and Fig. S4 C). Additionally, these embryos had centrosome and perinuclear regions that were densely labeled by Patronin:GFP, and these pools persisted throughout the GBE (Fig. 5, A–C and Fig. S4 C). Centrosomal Patronin:GFP increased by 4.3-fold and 3.8-fold at the onset and 20 min of GBE, respectively, as compared to control embryos (Fig. 5 B). The mean perinuclear Patronin:GFP intensity was increased similarly by 4.7-fold at the onset and 3-fold

at 20 min compared to control embryos (Fig. 5 C). We next tested if these increased Patronin levels were functional and if they altered the MT network. Remarkably, γ-tubulin compromised embryos exhibited enhanced centrosomal and perinuclear MTs (Fig. 5, D and E). We also observed that depletion of γ-tubulin led to an increase of 56% in MT intensities at regions apical to the nuclei as compared to the control (Fig. S4 D). Furthermore, co-depletion of Patronin and γ-tubulin showed a reduction of centrosomal MTs by ~30% as compared to γ-tubulin perturbed embryos (Fig. S4 E). This suggests that Patronin enrichment at centrosomes can powerfully compensate for the reduction in γ-tubulin function. Our measurements show an approximately 2-fold increase in both centrosomal and perinuclear MT intensities (2-fold centrosomal MT increase at both 0 and 20 min, and 1.9-fold and 1.7-fold perinuclear MT increase at 0 and 20 min GBE, respectively) compared to control intensities (Fig. 5, F and G). However, it appears that these enhanced intensities do not affect the behaviors of the nucleus-associated ncMT bundles, as a similar apical shifting of the perinuclear MTs occurs during GBE as in control embryos (Fig. 5, D and H). Similarly, we did not observe significant differences in the intensity of acetylated MTs in the apical and perinuclear regions (Fig. S4, F and F′).

We next wanted to test how nuclear positioning and dispersion are impacted when excess MT bundles are present in the embryonic epithelium. γ-tubulin embryos had a mild decrease in nuclear dispersion, with 67% of nuclei present in the top apical 10 μm of the cell, while control embryos had 56% of nuclei present in this region (Fig. 6, A and B). We did not observe nuclear invasion of the apical exclusion zone in these embryos indicative of largely normal nuclear positioning and MT-cortex interactions. Consistent with the mild dispersion defect, mean and peak nuclear speeds were also slightly decreased in these embryos (0.36 μm/min and 0.85 μm/min in γ-tub embryos as compared to 0.44 μm/min and 1.31 μm/min in control embryos) (Fig. 6, C and D), and we also observed a modest reduction in MSD-detected active nuclei (53%) as compared to control embryos (71%) (Fig. 6 E). These data indicate the existence of a Patronin/γ-tubulin antagonistic interaction and that Patronin recruitment to the centrosome can compensate, at the level of MT intensities, for γ-tubulin depletion.

## Compromising EB1 function disrupts a shift to apical MT networks

EB1 is a MT binding protein that has been previously studied during GBE—it is well-appreciated for its ability to mark +ends of MT and serve as a scaffolding protein (Mimori-Kiyosue et al., 2005; Ambrose and Wasteneys, 2008; Ambrose et al., 2011; Dugina et al., 2016; Garcia De Las Bayonas et al., 2019; Aher et al., 2020; Dema et al., 2022). We were therefore curious to examine its function in nuclear dispersion and MT dynamics. Disruption of EB1 revealed intense MT bundles around the nucleus (Fig. 7, A–C). Unlike in control embryos, these MT bundles persisted in close association with nuclei even as GBE continued (2-fold and 3-fold increase at 0 and 20 min as compared to control) and, intriguingly, there was little observable shift to apical regions (Fig. 7, A–D). Centrosomal MT intensities in *EB1* embryos were ~30% higher at GBE onset (Fig. 7 E) and acetylated MT immunostaining was near control levels (Fig. S5, A and A′).

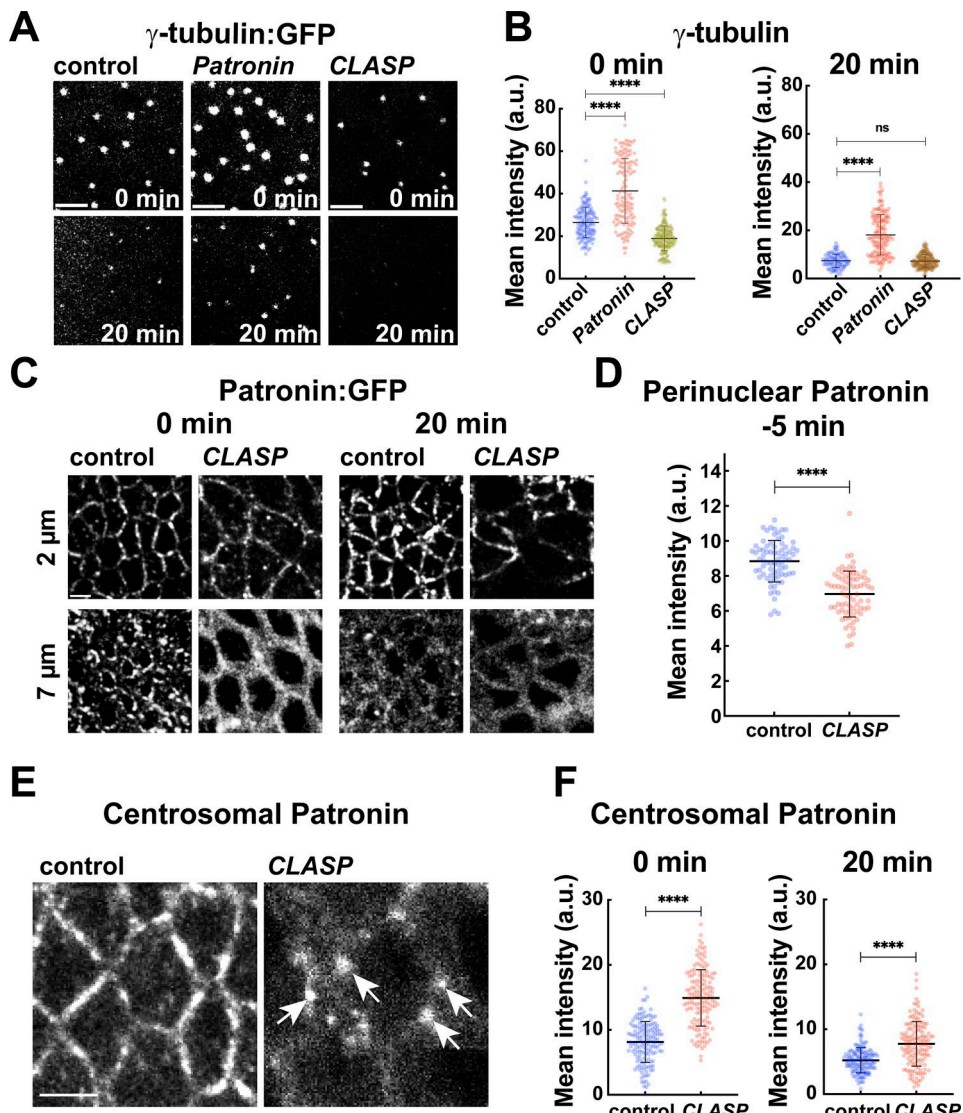

**Figure 4. Quantitation of centrosomal MTs in *CLASP* and *Patronin* embryos reveals an antagonistic relationship between centrosomal and ncMT networks. (A)** γ-tubulin:GFP intensities vary in control, *Patronin*, and *CLASP* embryos. **(B)** Quantification of γ-tubulin:GFP in control, *Patronin*, and *CLASP* embryos at 0 and 20 min GBE; $n = 150$ for control and *Patronin*, and 200 centrosomes for *CLASP* at 0 min and $n = 88$, 150 and 172 centrosomes for control, *Patronin* and *CLASP*, respectively, at 20 min; control and *Patronin* k = 3 and *CLASP* k = 4 embryos. **(C)** Still images showing the localization of Patronin:GFP in control and *CLASP* embryos at 0 and 20 min GBE, highlighting the absence of perinuclear Patronin in *CLASP* embryos. **(D)** Perinuclear Patronin:GFP intensities in control and *CLASP* embryos 5 min before GBE onset showing depleted perinuclear intensities in *CLASP* embryos; $n = 75$ perinuclear regions from k = 3 embryos for each background. **(E)** Still images showing Patronin:GFP enrichment at centrosomes (5 μm below cell apices) in *CLASP* embryos unlike in control embryos (same control as Fig. 5 A and Fig. 7 A, 0 min at 5 μm), arrows mark centrosomal Patronin. **(F)** Centrosomal Patronin:GFP intensities at 0 and 20 min show upregulated centrosomal Patronin in *CLASP* compared to the control embryos; $n = 150$ centrosomes and k = 3 embryos for each background. Scale bar = 5 μm. All scatter plots show the mean ± SD. Statistical significance was calculated using the Mann–Whitney U-test. ns, not significant. ****$P < 0.0001$.

Consistent with higher perinuclear MT intensities, Patronin:GFP was also increased near nuclei after EB1 disruption (Fig. 7, F and G; and Fig. S5 B). Thus, MTs stay strongly associated with nuclei and have a reduced ability to detach from nuclei and displace apically in embryos with compromised *EB1* function.

Given the observed persistence of nuclear MT bundles, we next assessed how nuclear behaviors are impacted during germband extension in *EB1* embryos. *EB1* embryos possessed an increase in the percentage of nuclei that were located apically (65% of nuclei are within 10 μm of the apical surface in *EB1* disrupted) (Fig. 7 H). *EB1* embryos also exhibited a mild increase

in the percentage of nuclei crowding into the apical exclusion zone (14%) as compared to control embryos (Fig. 7, C and I). These results suggest that nuclei possessing perinuclear MTs is not sufficient for dispersion, but that the transition of perinuclear MT bundles away from the nucleus is a key driver of nuclear dispersion. The collapse of nuclei against the apical cortex also indicates the potential loss of an MT-cortex interaction that is required for nuclear anchoring and positioning (Fig. 7, C, I, and J). We did not observe nuclei in *EB1* embryos that had large grooves on their apical surface, as was observed after *CLASP* disruption, demonstrating that persistent nuclear MTs can

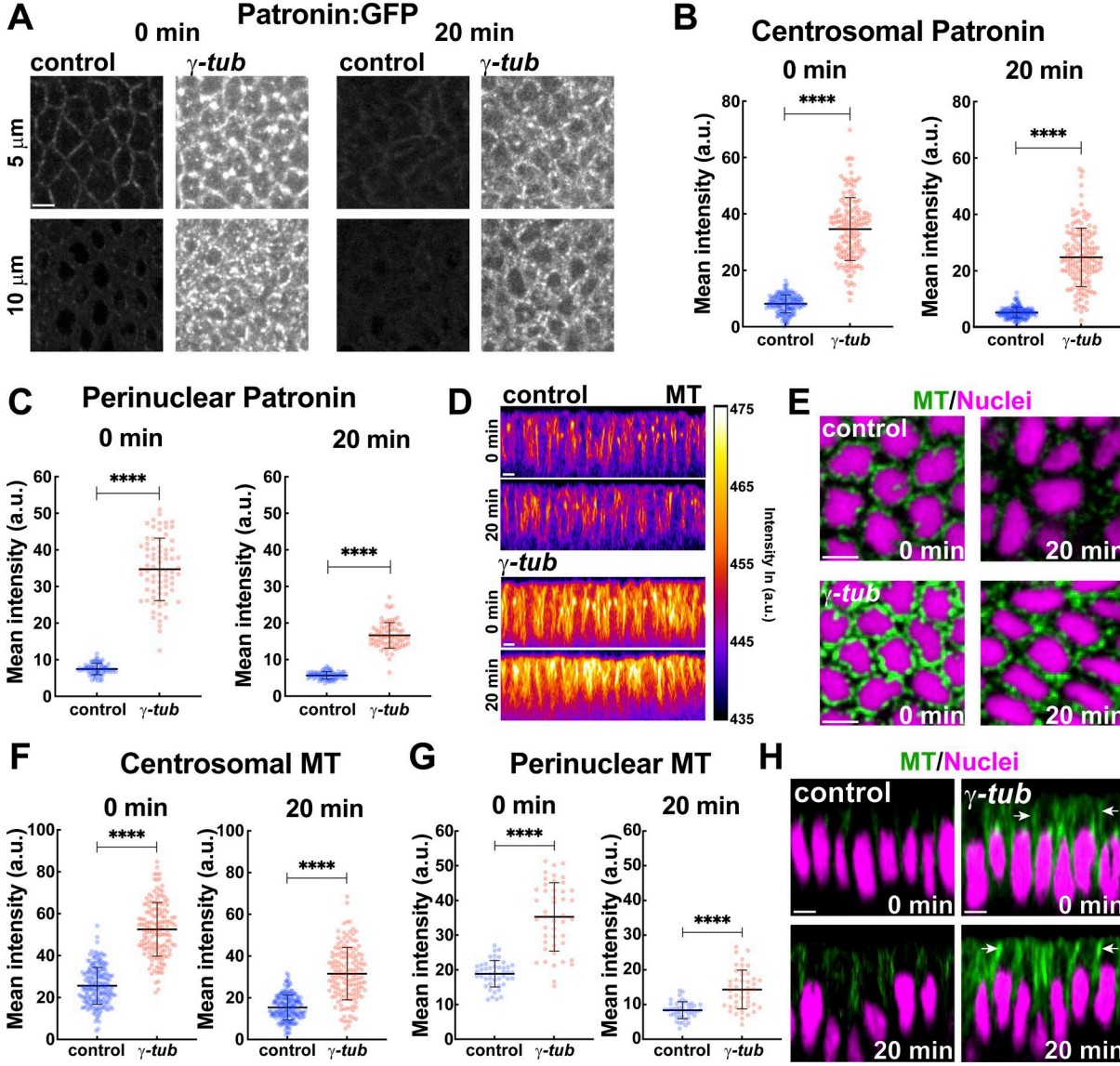

Figure 5. **γ-tubulin–disrupted embryos have enhanced Patronin intensities. (A)** Still frames revealing upregulated Patronin:GFP throughout the GBE in γ-tubulin37C shRNA (*γ-tub*) embryos as compared to control embryos. **(B)** Centrosomal Patronin:GFP intensities in control and *γ-tub* embryos; *n* = 150 centrosomal regions from *k* = 3 embryos for each background. **(C)** Perinuclear Patronin:GFP intensities in control and *γ-tub* embryos; *n* = 75 perinuclear regions from *k* = 3 embryos for each background. **(D)** Orthogonal view of MT natural log intensity heatmap showing enhanced MT bundles in *γ-tub* embryos compared to control. **(E)** Cross-section of MTs (green) and nuclei (magenta) showing enhanced and persistent perinuclear MT in *γ-tub* embryos as compared to control. **(F)** Centrosomal MT intensities in control and γ-tubulin37C embryos; *n* = 200 and 150 centrosomal regions for control and *γ-tub*, respectively, *k* = 4 for control and *k* = 3 *γ-tub* embryos. **(G)** Perinuclear MT intensities showing enriched MT pools in *γ-tub* perinuclear regions as compared to controls; *n* = 45 perinuclear regions *k* = 3 embryos for each background. **(H)** Orthogonal views of MT and nuclei showing MT bundles still shift apically in *γ-tub* embryos (arrows). Fig. 2 C; Fig. 3 C; Fig. 5, A and H; and Fig. 7, C and F control images/plots reproduced for comparison purposes. Scale bar = 5 µm. All scatter plots show the mean ± SD. Statistical significance was calculated using the Mann–Whitney U-test. ****P < 0.0001.

preserve the apical–basal orientation of nuclei. Consistent with *EB1* embryos having defects in the active transport of nuclei, only 41% of *EB1* nuclei had MSD active periods (as compared to 71% in control, 45.8% in *Patronin*, and 26.2% for *CLASP* embryos) (Fig. 2 J, Fig. 3 K, and Fig. 7 K). *EB1* nuclei also had peak speeds of 0.94 µm/min (compared to 1.31 µm/min in control) and average speeds of 0.37 µm/min (compared to 0.44 µm/min in control) (Fig. 7 L and Fig. S5 C). Thus, these observations suggest that EB1 may promote MT-cortex interactions and its loss disrupted the displacement of perinuclear MT networks from nuclei to form

the apically shifted MTs that lead to the dispersion and positioning of nuclei in the germband epithelium.

## Discussion

During GBE, nuclei are repositioned to different apical–basal planes to allow for maximal changes in cell shapes and the sliding of cellular volumes past neighboring cells for efficient cell intercalation and tissue extension (de Leeuw et al., 2024). Here, we have shown that a rapid reorganization of MT networks is

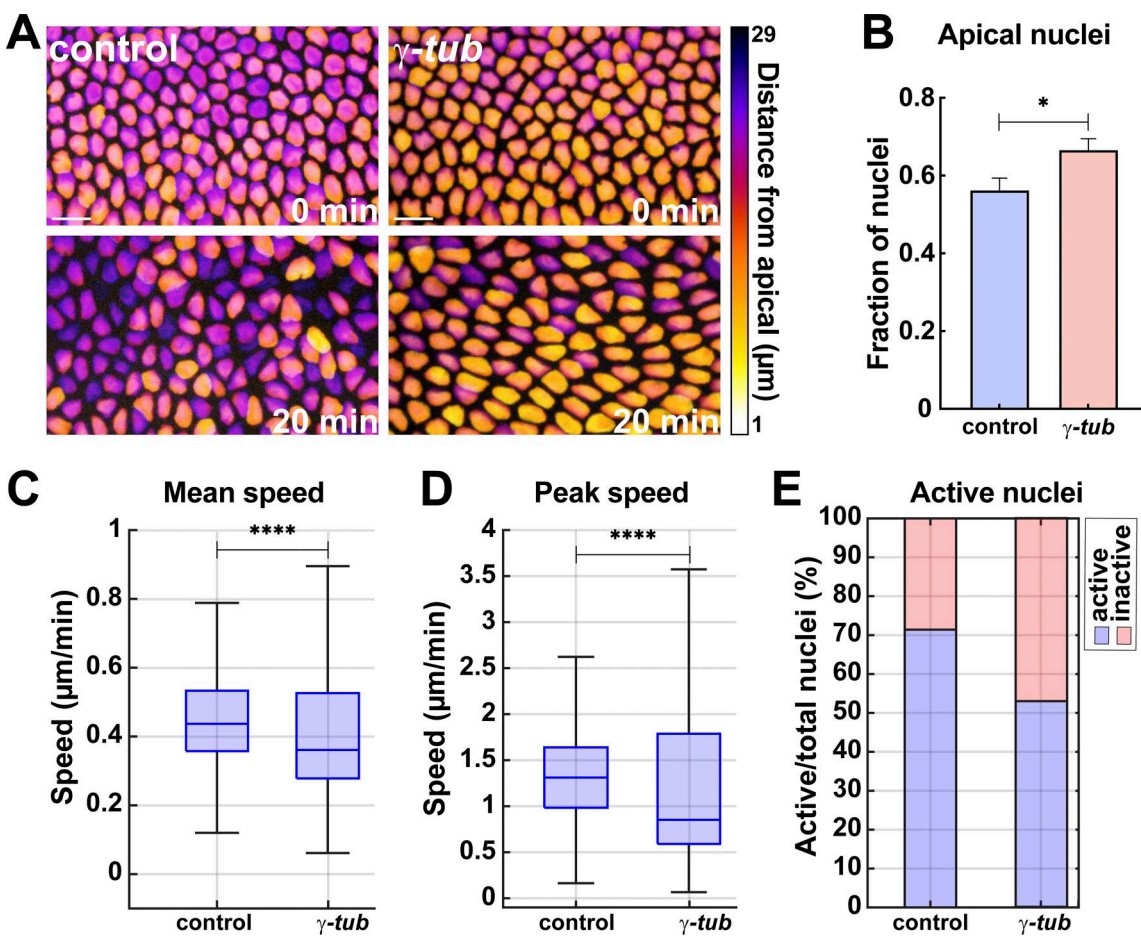

Figure 6. **Nuclear behaviors after γ-tub disruption. (A)** Nuclei color-coded for apical–basal position in control and γ-tub embryos. **(B)** Fraction of nuclei in apical 10 μm of cell in control and γ-tubulin embryos; $n$ = 263 and 273 nuclei for control and γ-tub, respectively, k = 3 embryos each background. **(C)** Mean nuclear speeds in control and γ-tub embryos. **(D)** Peak nuclear speeds in control and γ-tub embryos; $n$ = 546 and 612 in control and γ-tub, respectively, from k = 3 embryos for each background. **(E)** Percent of active nuclei as detected by MSD in control and γ-tub embryos. (C and E) $n$ = 546 and 1,223 nuclei in control and γ-tub, respectively, from k = 3 embryos for each background. Scale bar = 10 μm. Statistical significance was calculated using the Mann–Whitney U-test. *$P$ < 0.05, ****$P$ < 0.0001.

necessary for these nuclear movements within the epithelial cytoplasm. MTs transition from a largely centrosomal organization to perinuclear and apical non-centrosomal arrays that are required for the proper dispersion of nuclei away from a common apical plane (model, Fig. 7 M). The centrosomal to ncMT switch begins during mid-late cellularization when MTs initiating from centrosomes lose contact with the centrosome (Harris and Peifer, 2007). Our work illustrates that as GBE advances, the MT basket that initially surrounds the nucleus shifts towards apical cell regions in only 20 min of developmental time (Fig. 7 M). These nucleus-associated MTs have attributes of a stabilized ncMT array, as judged by the appearance of acetylated MTs and FRAP data. CAMSAP (Patronin) and +TIP CLASP protein activity is essential for the stabilization of this network, both before and during the apical shift away from nuclei. This is consistent with previous work that demonstrated that Patronin overexpression enhanced tyrosinated and acetylated MTs in the apical domes of early embryos (Takeda et al., 2018).

Interestingly, CLASP function additionally appears to mediate the orientation and anchoring of nuclei during dispersion, as

they begin to wobble and collide with apical surfaces after CLASP disruption. The destabilization of perinuclear MTs seen in CLASP and Patronin embryos is not apparent in EB1 compromised embryos, suggesting that this +TIP protein is not required to stabilize the ncMTs. However, perinuclear MTs fail to migrate away from nuclei, perhaps indicating that EB1 is needed to associate these MTs with the cortex, which would be consistent with canonical cortical functions of EB1 (Mimori-Kiyosue et al., 2000; Mimori-Kiyosue et al., 2005; Askham et al., 2000; Barth et al., 2002). These effects of Patronin, CLASP, and EB1 function are also apparent at the level of metrics that report nuclear velocities and active periods of movement. The percentage of actively dispersing nuclei drops sharply in embryos with disrupted CLASP, Patronin, and EB1 function (reductions of 65%, 38%, and 43%, respectively). Similarly, individual nuclear velocities decreased by 27–50% in Patronin, CLASP, and EB1 embryos. Thus, a stable ncMT network, created and positioned through the combined activities of –TIP and +TIP proteins, is an essential mediator of dispersive activities in the intercalating epithelium.

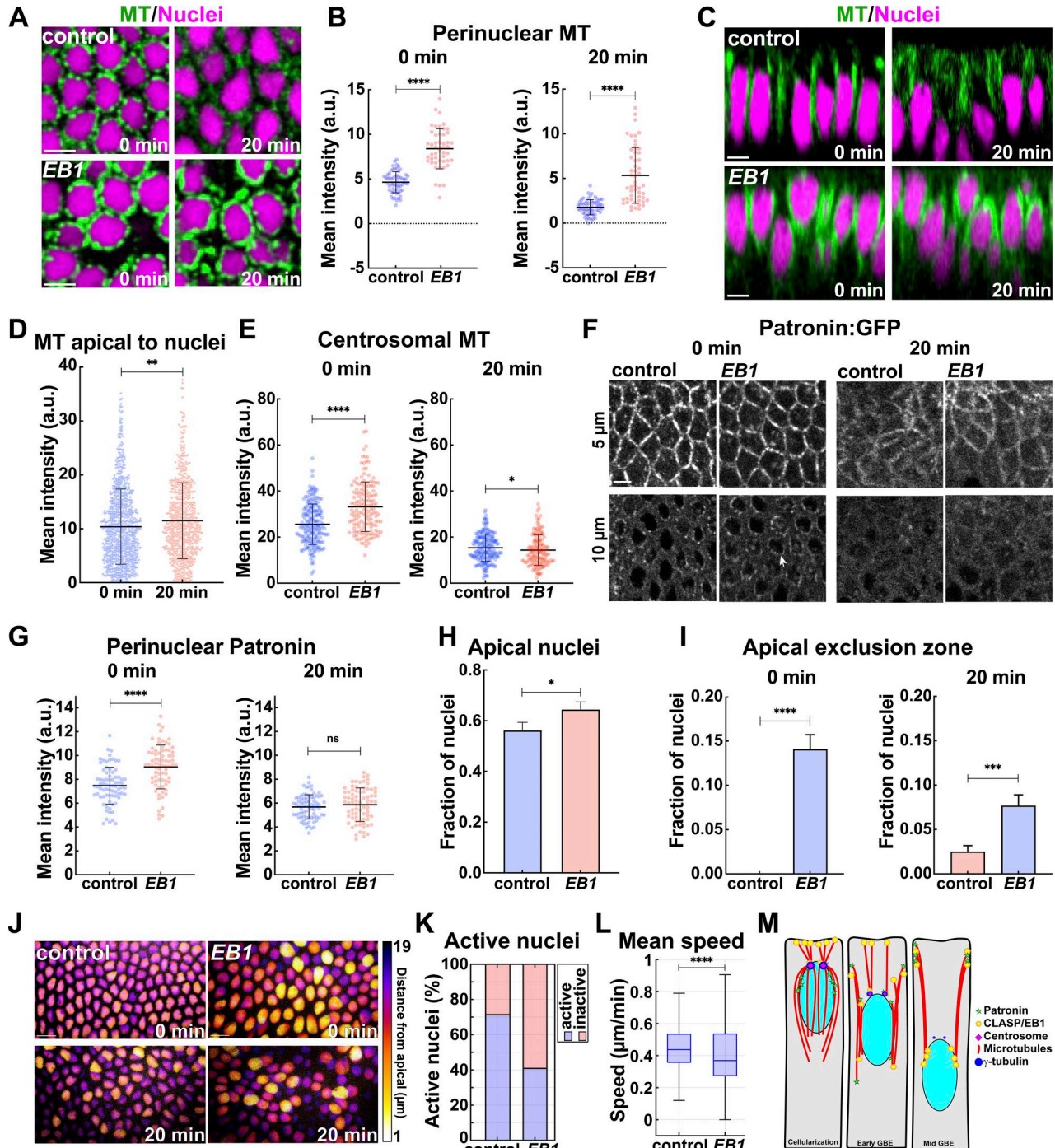

Figure 7. **EB1 function is required for nuclear dispersion and the ncMT shift towards apical regions. (A)** Cross-section of MTs (green) and nuclei (magenta) showing enhanced and persistent perinuclear MTs in EB1 shRNA (*EB1*) embryos compared to control. **(B)** Perinuclear MT intensity measurements showing enriched MT pools in perinuclear regions in *EB1* embryos; n = 60 and 48 for control and *EB1*, respectively, at 0 min, and n = 61 and 52 perinuclear regions in control and *EB1*, respectively, at 20 min from k = 3 embryos for each background. **(C)** Orthogonal views of MT and nuclei showing perinuclear MT bundles fail to shift apically in *EB1* embryos. **(D)** Measurement of apical MT intensities at 0 and 20 min suggesting a failure of apical MT enrichment in *EB1* embryos; n = 796 and 614 apical regions for 0 and 20 min, respectively, from k = 3 embryos. **(E)** Centrosomal MT intensities are enhanced when *EB1* function is compromised; n = 200 and 150 centrosomal regions for control and *EB1*, respectively, from k = 4 embryos for control and k = 3 embryos for *EB1*. **(F)** Enhanced perinuclear Patronin: GFP in *EB1* embryos (arrow) as compared to control embryos. **(G)** Perinuclear Patronin:GFP intensities are enhanced in *EB1* embryos as compared to control embryos; n = 75 perinuclear regions from k = 3 embryos for each background. **(H)** Fraction of nuclei in the apical 10 µm of the cell in control and *EB1* embryos. Error bars indicate the standard error of mean; n = 263 and 288 nuclei for control and *EB1*, respectively, from k = 3 embryos for each background. **(I)** Fraction of nuclei invading the apical exclusion zone in control and *EB1* embryos; n = 455 and 434 nuclei in control and *EB1*, respectively, at 0 min and n = 483 and 469 nuclei in control and *EB1*, respectively, at 20 min from k = 3 embryos for each background. **(J)** Maximum-intensity projections of nuclei color coded for distance from cell apices in control and *EB1* embryos. **(K)** Percent of active nuclei as detected by MSD. **(L)** Mean nuclear speeds in control and *EB1* embryos. **(K and L)**, n = 546 and 383 nuclei in control and *EB1*, respectively, from k = 3 embryos for each background. **(M)** Model of MT network transitions during GBE, showing that a

centrosomal MT network transitions to a ncMT network during GBE with the aid of EB1, CLASP, and Patronin function while nuclei are driven into deeper cell regions. Scale bar = 10 µm for (J) and 5 µm for (A, C, and F). Fig. 2 C; Fig. 3 C; Fig. 5, A and H; Fig. 7, C and F; and Fig. 4 E control images/plots reproduced for comparison purposes. All scatter plots show the mean ± SD. Statistical significance was calculated using the Mann–Whitney U-test. ns, not significant. *P < 0.05, **P < 0.01, ***P < 0.001, and ****P < 0.0001.

Our results also suggest an intriguing antagonism between centrosomal and ncMTs, as well as components of the machinery that generate these MT arrays, at these stages. Patronin or CLASP disruption, which weakened ncMT populations, produced exceptionally bright centrosomes of increased MT intensities. Conversely, depleting γ-tubulin caused an increase in apical and perinuclear MTs. This antagonism could be through a competition for tubulin subunits, such that the absence of ncMT or centrosomal arrays increases the availability of subunits for the competing population. However, this antagonism is also apparent at the level of the stabilizing proteins. γ-tubulin depletion permitted the enhanced recruitment of Patronin to centrosomes and, similarly, the disruption of CLASP function also saw an increase of centrosomal Patronin. This may explain a surprising result—centrosomal MT intensities actually increased when γ-tubulin function, one of the key proteins in nucleating centrosomal MTs, is compromised. The enhanced recruitment of Patronin to centrosomes observed after γ-tubulin depletion (and the reversal of MT enhancement observed after Patronin/γ-tubulin co-depletion) suggests that Patronin can substitute for weakened γ-tubulin activity, thus allowing the stabilization of centrosome-associated MTs. It is interesting to note that compensatory mechanisms for MT organization during mitotic divisions have also been observed, although these have implicated pathways involving Msps, mei-38, or augmin (Hayward et al., 2014; Zhu et al., 2023). These results thus highlight a dynamic interplay between ncMTs and centrosomal MTs.

Lastly, how are these ncMT networks seeded? Previous work has shown that ncMT networks are sometimes formed through a "release-and-capture" mechanism in which centrosomal MTs are severed by spastin/katanin family function and then stabilized and repositioned as ncMT arrays (Mogensen, 1999; Abal et al., 2002; Brodu et al., 2010; Jiang et al., 2014; Sanchez and Feldman, 2017; Tillery et al., 2018; McNally and Roll-Mecak, 2018; Takeda et al., 2018; Gillard et al., 2021; Kuo and Howard, 2021). Although we have yet to observe defects at these stages when these family members are disrupted, it may be significant that Patronin was observed in a juxta-centrosomal location, where it could mediate such a handoff. Additionally, centrosomal function is rapidly diminished during GBE—this downregulation could result in the freeing of centrosomal MTs for Patronin-mediated stabilization and eventual recruitment to the perinuclear and apical MT arrays independent of a severing function. Centrosomal deactivation is a common feature in many different epithelial systems (Mogensen and Tucker, 1987; Müsch, 2004; Brodu et al., 2010; Feldman and Priess, 2012). The reasons for this are not always clear, but it may be that this downregulation permits the construction of robust ncMT networks essential for secretion, transcytosis, lumen formation, or (as observed here) nuclear migration and positioning (Starr, 2017; Gimpel et al., 2017; Tillery et al., 2018; Zheng et al., 2020; Ricolo and Araujo, 2020; Gillard et al., 2021).

# Materials and methods

## Fly stocks and genetics

Fly stocks used in this study are ubi:RFP-NLS (#34500 and #30555; BDSC), UAS:white Val20 (#33623; BDSC; HMS00017; Kotov et al., 2020), UAS:γ-tub37C Val20 (#32513; BDSC; Colombié et al., 2013), UAS:CLASP Val20 (#34699; BDSC; HMS01146; Barlan et al., 2017), UAS:EB1 Val22 (#36599; BDSC; GL00559; Kim et al., 2024), UAS:Patronin Val20 (HMS01547; Gillard et al., 2021; Morton et al., 2025), Jupiter-GFP (#3686; BDSC), Ubi:Patronin-GFP (#55129; BDSC), ncd:γ-tubulin37C-GFP (#57328; BDSC), Gap43-mCh, Asl-mCh (Conduit et al., 2015) (Jordan Raff, University of Oxford, Oxford, England), matαTub-Gal4VP16-67C;15 (D. St. Johnson, Gurdon Institute, Cambridge, UK), Patronin WALIUM 22 (this study), and UAS GFP-CLASP (this study). Patronin WALIUM 22 shRNA was constructed using the following primers: forward, 5′-CTAGCAGTGGCTCAAGCTCGAATCTAA TAGTTATATTCAAGCATATTAGATTCGAGCTTGAGCCGCG-3′; and reverse, 5′-AATTCGCGGCTCAAGCTCGAATCTAATATG CTTGAATATAACTATTAGATTCGAGCTTGAGCCACTG-3′. The forward and reverse primers were annealed, cloned into pWA-LIUM22 vector, sequenced and then sent to BestGene for injection. For UAS GFP-CLASP construct, eGFP was amplified with the following primers: forward, 5′-AAGGTACCATGGTGAGCAAGGGCG AG-3′ and reverse, 5′-AAACTAGTGCTAGCCTTGTACAGCTCG TCCATGCC-3′. The amplified eGFP was inserted into the UASp vector using KpnI and SpeI restriction enzymes. The NheI restriction recognition site was introduced to the reverse primer of GFP for the insertion of the CLASP gene. CLASP cDNA (#LD11488) was obtained from the Drosophila Genomic Resource Center and was amplified using the following primers: forward, 5′-AAAGCT AGCGCCTATCGGAAGCCCAGCG-3′; and reverse, 5′-AAAGGATCC TCATGACGACGATGCCGCG-3′. The amplified product was inserted using NheI and BamHI restriction enzymes. All reagents for the molecular cloning were obtained from New England Biolabs. Sequence of the resulting construct was confirmed, and construct was injected by BestGene, Inc. UAS transgenic flies with the desired markers were crossed with matαTub-Gal4VP16-67C driver females and second-generation embryos were used for the study. We used FlyBase (FB2020 to FB2025) for information on genes, phenotypes, function, stocks, gene expression, and more.

## Microscopy, live imaging, and injections

Embryos were collected on yeasted apple juice agar plates, dechorionated in 50% bleach solution for 2 min, washed, and then transferred to an air-permeable membrane. The embryos were covered with Halocarbon 27 oil (Sigma-Aldrich), and a coverslip was placed on the embryos for live imaging. All live imaging was performed at 25°C on either a CSU-W1 Yokogawa spinning-disk confocal from Zeiss/3i Intelligent Imaging Innovations with a 63× 1.3 NA objective captured with a Prime 95 sCMOS camera using Slidebook software or a CSU10b Yokogawa spinning-disk

confocal from Zeiss/Solamere Technologies Group with a 63× 1.4 NA objective captured with a Hamamatsu ORCA EMCCD or Prime 95 sCMOS camera using Micro-Manager software.

For quantification of centrosomal MTs, time-lapse movies with 31 z-stacks with 1 μm spacing were taken every 5 min to minimize photobleaching. For perinuclear MT measurements, time-lapse movies of 20 z-slices at 1 μm spacing every 17 s intervals or every 5 min to minimize photobleaching for quantitation were taken. For centrosomal and perinuclear Patronin measurements, time-lapse movies with 31 z-slices with 0.5 μm spacing and 20 s intervals were taken. For γ-tubulin intensity measurements, movies were taken at 1 μm z-slices (20 slices) every 20 s.

For injection experiments, embryos were dechorionated as described above, placed in apple juice agar, staged, glued to the coverslip, and dehydrated for 12 min. Then the embryos were covered with Halocarbon 700 oil and injected with Colchicine (cat# C3915; 2 mg/ml in water; Sigma-Aldrich) and time-lapse imaging was performed with acquisition of 31 z-slices at 1 μm spacing and 20 s intervals.

### Embryo fixation, antibodies, immunostaining, and imaging
Embryos were dechorionated in 50% bleach, washed, and then immersed in heptane. A mixture of Paraformaldehyde (32%; Electron Microscopy Services) and EDTA (final concentration of 12.8% PFA and 4 mM EDTA) was slowly added to the immersed embryos under gentle vortexing. The embryos were incubated in the fixative solution for 15 min with horizontal shaking and then manually devitellinized. Antibodies used for immunostaining are mouse anti-acetylated α-tubulin (1:4,000; T7451; Millipore Sigma), rabbit anti-GFP antibody (1:500; A-11122; Invitrogen), Alexa Fluor 488 goat anti-rabbit antibody (1:500; A-11034; Invitrogen), Alexa Fluor 647 phalloidin (1:500; A-22287; Invitrogen), and Alexa Fluor 568 goat anti-mouse antibody (1:500; A-11031; Invitrogen). Embryos were mounted in Prolong Gold Antifade reagent with DAPI (P36935; Invitrogen). Images were taken using an Olympus Fluoview FV1000 confocal laser-scanning microscope with a 40×/1.3NA objective.

### Nuclear segmentation
Prior to segmenting, raw image volumes were resampled and interpolated in Z using the MATLAB function imresize3, such that the dimensions of each voxel were equivalent in all directions. Image volumes were also smoothed with a 3D Gaussian filter in preprocessing to help reduce noise. Local thresholding was performed to generate rough nuclear segmentations, followed by a morphological erosion to transform those rough segmentations into seeds. Seeds were manually edited to separate any fused nuclei and to manually insert seeds as needed. These seeds were then used as imposed minima in a 3D watershed transform performed on the gradient of the resampled image volume as the edges of the nucleus correspond to the highest change in fluorescent intensity. Nuclei were given unique tracking IDs post-segmentation that were validated by comparing maximal overlapping objects over time.

### Nuclear midplane identification and tracking
All NLS-labeled data consist of an imaging volume that spans 20 μm in depth, and apical–basal nuclear positions were tracked by our defined the nuclear midplane (de Leeuw et al., 2024). Since nuclei in our system are often top-heavy, with more of their volume positioned apically than basally, using the widest portion of the nucleus is not ideal for determining nuclear midplanes. Thus, we define the nuclear midplane as the midpoint between the two half-max flanks of a given nucleus. Nuclei without a trackable maximized cross-sectional area were excluded from analysis.

### Active nuclear movement identification
We used a rolling analysis window technique to detect periods of active motion in each nuclear trajectory, adapted from previous studies (Huet et al., 2006). MSD is a customary method to identify periods of active, diffusive, or constrained motions, based on whether the MSD curves upwards, is linear, or curves downward, respectively. For periods of active motion, the MSD behaves as a power law $MSD(\tau) \propto \tau\gamma$, where $\gamma > 1$. We calculated the parameter gamma along each nucleus trajectory, fitting the MSD to lags between 4 and $3(N-1)/4$, where $N$ is the odd number of points in the window. We excluded the first 3 lags from the fitting to correct for artifacts at short time scales. Gamma fitting was performed by applying a linear fit to a log-log plot of the MSD versus $\tau$, to reduce computation time. Trajectories were filtered with a fifth-order median filter (using the *medfilt1* function in MATLAB) to remove noise.

### Nuclear speeds
We tracked the midplane position of each nucleus relative to the apical-most cell surface z-layer over time and calculated the apical–basal velocities over a 1-min time window for all genotypes. In the included peak and average velocity plots, each tracked nucleus contributes a single data point, representing its max/mean velocity, respectively. All box and whisker plots represent 25th quartile (bottom of the box), median (mid of the box), and 75% quartile (top of the box), and the whiskers represent the minimum (below the box) and the maximum (above the box) values.

### Image processing, editing, and figure preparation
Images acquired from spinning-disk and laser-scanning confocal microscope were processed in FIJI using the Difference of Gaussian technique. The Gaussian filter was applied to reduce noisy imaging. Wherever time and/or background comparisons were made, images were identically leveled. The images were edited and prepared using Adobe Photoshop and Illustrator. All scatter plots, line scan plots and bar graphs were generated using GraphPad Prism.

### Orthogonal crops with line scan profile
Orthogonal images at different time points were obtained from time-lapse 3D movies. The maximum-intensity value from the background (signals above the cells) was measured and subtracted from image values. A Gaussian blur of sigma 1 was applied to the images. Using a line tool, an ROI of width 3 pixels was drawn at 2 μm above the nuclear surface or at the nuclear midplane, the mean intensity was measured using FIJI. The values were plotted using GraphPad Prism (Fig. 1, A′′–C′′′).

## Intensity heatmaps

Intensity heatmaps for Patronin:GFP and MT were produced by taking the orthogonal section from the movies, leveled identically between controls and knockdowns, and heatmaps were generated using Fire LUT in FIJI. For Fig. 5 D, the intensity heatmaps were generated by natural log transformation of the raw images of orthogonal projections, then using Fire LUT and leveling identically with controls in FIJI.

## Color-coded heatmaps for nuclei

Unique color at each z-steps were assigned to nuclei using the FIJI plugin z-stack Depth Colorcode 0.0.2 (https://github.com/UU-cellbiology/ZstackDepthColorCode). Then maximum projection of the obtained output was generated to color-code the nuclei based on their position from the apical.

## Fluorescence intensity measurements

### MT intensity

Time-lapse 3D images obtained from the spinning-disk confocal microscope or fixed and immunostained embryos obtained from the laser-scanning confocal microscope were used. For intensity comparisons between time or backgrounds, the embryos were imaged and quantified using identical parameters.

Centrosomal MT intensity measurements were obtained from embryos expressing Asl:mCh and Jupiter:GFP in different perturbation backgrounds. Using the circle tool in FIJI, an ROI of 2.02 μm diameter was drawn around the Asl:mCh signal, and the mean intensity for Jupiter:GFP at the same spot was measured. The average background intensity (mean intensity measured from areas lacking Jupiter:GFP signal) was subtracted from mean intensities. Normalized intensity was calculated by subtracting the average background intensity from the mean intensity and dividing the value by the average mean intensity at 0 min. For all intensity measurements, outliers were identified and removed using the ROUT test with Q = 1% from these values in GraphPad Prism, and the resultant values were used to produce scatter plots (mean ± SD).

For perinuclear MT intensity, the freehand tool with 3-pixel thickness was used to draw an ROI outlining the nucleus 7 μm below the apical of the nucleus, and intensity for Jupiter:GFP was measured. For the background, the intensity from an ROI that did not include MTs was measured. The mean intensity plotted in the graph represents the background-subtracted mean intensity of the perinuclear region. Normalized intensity was calculated by dividing the background-subtracted mean intensity divided by the average mean intensity at 0 min.

Apical MT intensities were measured from orthogonal projections obtained from 0 to 20 min GBE. 1 × 5 μm (or 1 × 2 μm for Patronin, CLASP, and EB1 backgrounds) rectangular ROIs were drawn in the lateral regions apical to the nucleus. Mean intensities of Jupiter:GFP in each ROI were calculated. For background, the same size ROI was drawn in a region that did not include MTs, and the mean intensity from the Jupiter:GFP channel was calculated and averaged. The mean intensity obtained from the apicolateral regions was subtracted from the average background intensity and plotted.

## Perinuclear Patronin intensity

Perinuclear Patronin intensity was calculated from embryos expressing RFP-NLS and Patronin:GFP in different knockdown backgrounds. The movie was corrected for photobleaching using the exponential fit method from the bleach correction tool in FIJI and was used for subsequent measurements. Using the freehand tool with a 3-pixel width, an ROI was drawn around the nuclear periphery 4 μm below the apical of the nucleus. For the background intensity measurement, an ROI was drawn in a region that did not include Patronin particles. The mean intensity for perinuclear Patronin was calculated by subtracting the background mean intensity from the mean intensity of the perinuclear ROI.

## Centrosomal Patronin intensity

Embryos in different knockdown backgrounds expressing Asl:mCh and Patronin:GFP were used for this measurement. Asl:mCh was used to mark the centrosomes. A circle of diameter 2.02 μm was drawn around the centrosomes using the Asl marker and the mean intensity of Patronin:GFP was measured at that spot. The background intensity was measured from the ROI drawn in the region lacking Patronin:GFP signal. The mean intensity of Patronin:GFP was calculated by subtracting the background intensity.

## γ-Tubulin intensity

Two ROIs of diameter 15 pixels (small ROI, 0.168 μm/pixel) and 25 pixels (large ROI, which includes cytoplasmic area adjacent to centrosome) were drawn around the centrosome using the circle tool. The integrated density from both ROIs was measured. To calculate the background values, the integrated density of a small ROI was subtracted from large ROI. Then this value was converted to the mean intensity and subtracted from the mean intensity value of the small ROI.

## Fluorescence recovery after photobleaching

Photobleaching and imaging was performed on an Olympus Fluoview FV3000 confocal laser-scanning microscope with a 40×, 1.3 NA objective lens. The circle tool was used to draw ROI at the centrosomes or the perinuclear regions. One data point each for centrosomes and perinuclear regions was taken per embryo. Images were acquired every 1 s at 2 ms/px exposure for 60 s. The intensity measurements were performed in FIJI. For focal plane or biological drift correction, we normalized the data with the intensity from two non-photobleached regions. Normalized intensity ($N$) was calculated as:

$$N = \frac{(Ip - b) \times Ap}{(Inp - b) \times Anp}$$

where, $Ip$ = Intensity at photobleached region, $Ap$ = Area of photobleached region, $Inp$ = Intensity at non-photobleached region, $Anp$ = Area of non-photobleached region, $b$ = Background intensity.

The t50 and immobile fractions were calculated by fitting the FRAP data to one-phase association in Graphpad Prism. Immobile fractions were calculated by subtracting the plateau value from the normalized intensity measured before photobleaching.

### Measuring nuclei in the apical 10 µm of cells and in the apical exclusion zone

For each time point, each nucleus was measured for the distance from the cell apices. The nuclear population was then assigned to three different categories based on these distances to (1) residing in the apical-most 2 µm, (2) residing in the apical 10 µm, or (3) residing below the apical 10 µm of cell volumes. Since the apical–basal height of the cells at germband extension stage was ~30–35 µm, we decided to choose 10 µm as it represents the apical 1/3$^{rd}$ region of the cell. The fraction for each of the categories was calculated by dividing the population in each category by the total nuclear population counted at each time point.

### Online supplemental material

There are 5 supplemental figures and 3 supplemental movies. Fig. S1 provides additional data demonstrating that perinuclear MTs are more stable than centrosomal networks. Fig. S2 illustrates the effects of *Patronin* disruption on MT arrays during GBE. Fig. S3 demonstrates that *CLASP* disruption destabilizes ncMT arrays. Fig. S4 depicts the impact of *γ-tubulin37C* disruption on Patronin localization and MT networks. Fig. S5 shows that *EB1* disruption affects MT networks associated with apical cell regions. Video 1 shows how MTs remodel during the course of GBE. Video 2 demonstrates the effects of *Patronin* disruption on MTs. Video 3 shows that centrosomal MTs are enriched after *CLASP* disruption.

### Data availability

All measurements include imaging data from at least three embryos. Statistical significance was calculated with a Mann–Whitney U-test. ns = not significant; *P < 0.05; **P < 0.01; ***P < 0.001; and ****P < 0.0001. Error bars indicate standard error of mean (SEM). All primary data are freely available from the corresponding author upon reasonable request.

## Acknowledgments

We thank members of the Blankenship and Loerke labs for critical reading and constructive comments on the manuscript.

This work was supported by grants from the NIH NIGMS: R01GM127447 to J. Todd Blankenship and Dinah Loerke. Stocks obtained from the Bloomington Drosophila Stock Center (NIH P40OD018537) were used in this study and are gratefully acknowledged.

Author contributions: Rashmi Budhathoki: conceptualization, data curation, formal analysis, investigation, methodology, project administration, resources, software, supervision, validation, visualization, and writing—original draft, review, and editing. Liam J. Russell: data curation, formal analysis, methodology, and software. Dinah Loerke: conceptualization, funding acquisition, project administration, and supervision. J. Todd Blankenship: conceptualization, data curation, formal analysis, funding acquisition, investigation, methodology, project administration, resources, supervision, validation, visualization, and writing—original draft, review, and editing.

Disclosures: The authors declare no competing interests exist.

Submitted: 16 July 2025

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

## Supplemental material

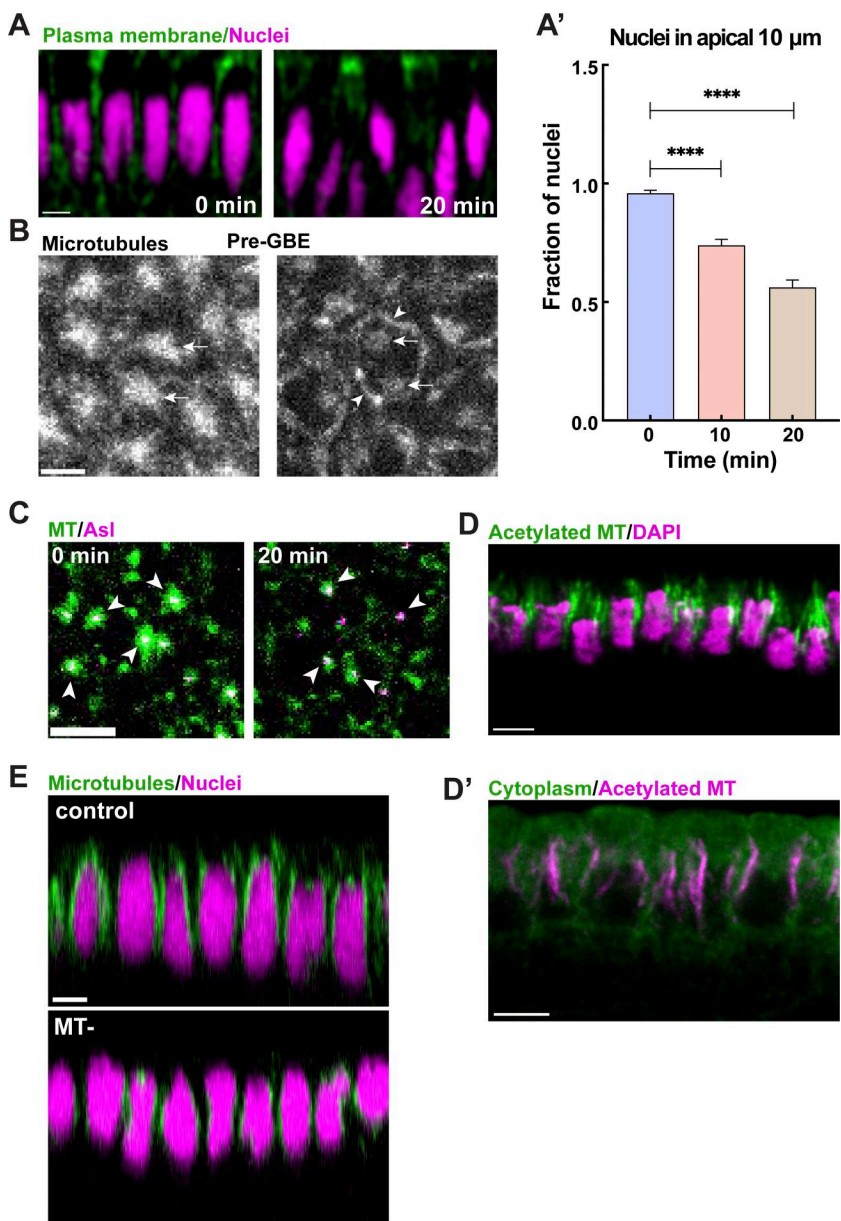

Figure S1.   **Perinuclear MT networks are more stable than centrosomal MT pools. (A)** Orthogonal projection of cells and nuclei at 0 and 20 min GBE, highlighting nuclear dispersion at later time points. **(A′)** Measurement of the fraction of nuclei in the apical 10 µm of cells showing the displacement of apically located nuclei to more basal regions as GBE proceeds (mean ± SEM); $n$ = 306, 355, and 263 nuclei at 0, 10, and 20 min, respectively. **(B)** Still frames from live imaging MTs (Jupiter:GFP) prior to GBE (cellularization) showing MTs transitioning from a centrosomal to nc networks. Arrows and arrowheads indicate centrosomal and nc MT pools, respectively. **(C)** Still frames showing MTs (green) and centrosomes (Asl, magenta) indicating the depletion of centrosomal MT pool at 20 min into GBE. Arrowheads point to centrosomal MTs. **(D)** Orthogonal projection of acetylated MT (green) and nuclei (DAPI, magenta), highlighting stable MTs in apical nuclear regions during GBE. **(D′)** Orthogonal projection showing acetylated MT (magenta) is present at perinuclear and juxtanuclear MTs. **(E)** Orthogonal projection of MTs (green) and nuclei (magenta) from embryos injected with vehicle control (top) and colchicine (bottom), showing colchicine-resistant perinuclear MT pools. Scale bar = 5 µm for (A, C, D, D′, and E), and 3 µm for (B). Statistical significance was calculated using the Mann–Whitney U-test. ****$P < 0.0001$.

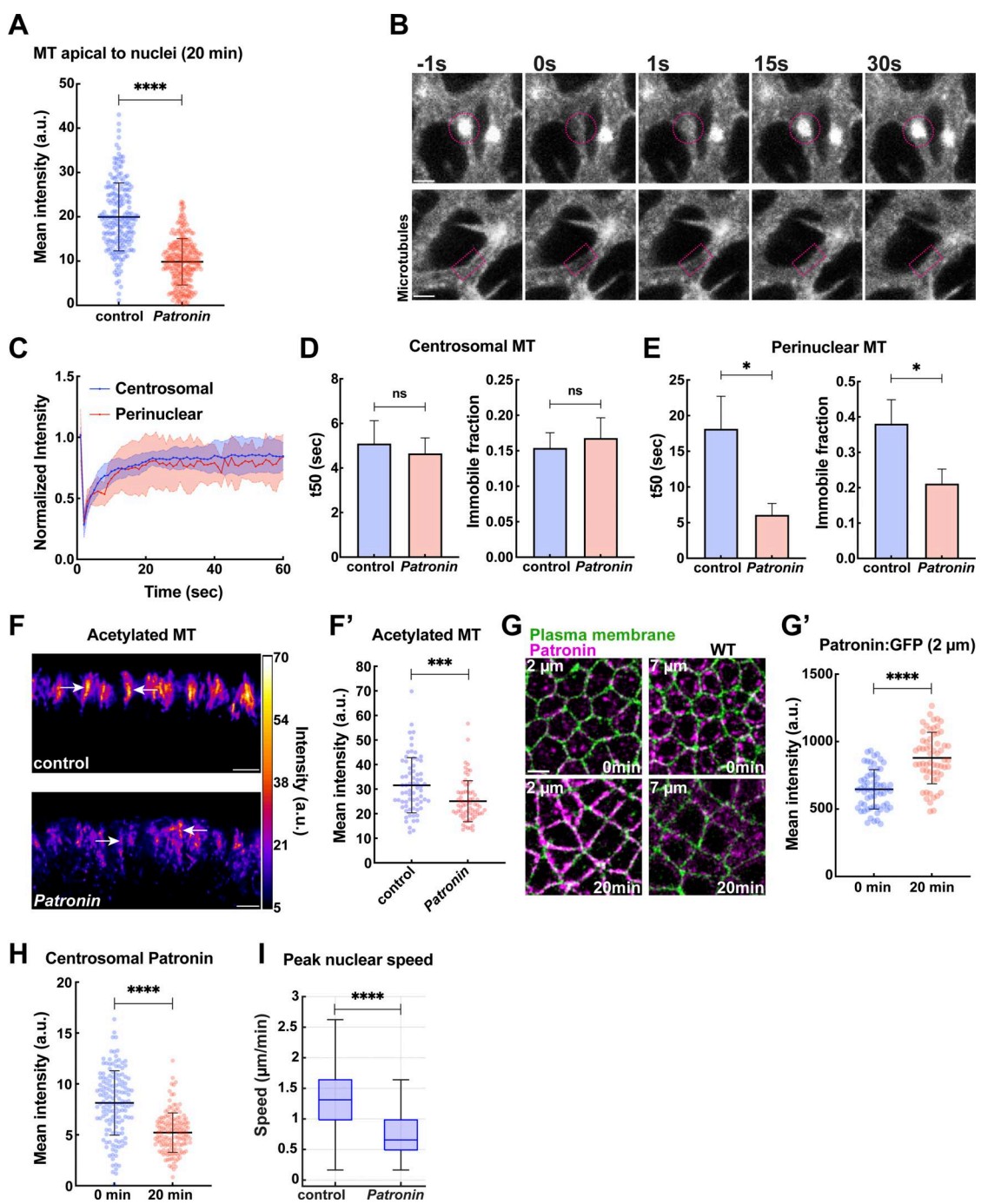

Figure S2. **Patronin function contributes to the stabilization of perinuclear MT networks. (A)** Quantification showing depleted MT intensities in apical regions above nuclei in *Patronin*-compromised embryos as compared to the control at 20 min into GBE; *n* = 200 regions for control and *n* = 287 regions for *Patronin* from k = 3 embryos for each background. **(B)** Still frames showing FRAP of α-tubulin:GFP at centrosomes (top) and perinuclear region (bottom) in *Patronin* embryos. Time is indicated in seconds; photobleaching was performed at 0 s. A circle or a rectangle in magenta indicates the photobleached region. **(C)** Fluorescence recovery profile for centrosome and perinuclear regions in *Patronin* embryos; *n* = 11 centrosomal regions from k = 11 embryos and *n* = 9 perinuclear regions from k = 9 embryos. **(D)** Halftime of recovery and immobile fraction for centrosomal α-tubulin:GFP in control and *Patronin* embryos (mean ± SEM); *n* = 11 centrosomal regions from k = 11 embryos for each background. **(E)** Halftime of recovery and immobile fraction for perinuclear α-tubulin:GFP in control and *Patronin* embryos (mean ± SEM); *n* = 11 and 9 perinuclear regions from k = 11 and 9 for control and *Patronin*, respectively. **(F)** Intensity heatmap of acetylated MT from immunostained control and *Patronin* embryos, revealing fragmented and destabilized MT at perinuclear regions when Patronin function is compromised. Arrows mark perinuclear acetylated MT. **(F')** Quantitation of acetylated MT pools; *n* = 70 regions from k = 7 embryos from each background. **(G)** Patronin (magenta) localization transitions during GBE (0 and 20 min). **(G')** Cortical Patronin:GFP intensities in region 2 μm below the apical cell surface at the onset and 20 min into GBE; *n* = 60 cells from k = 3 embryos for each background. **(H)** Patronin intensities at centrosomes at 0 and 20 min GBE; *n* = 150 centrosomal regions for each time point from k = 3 embryos. **(I)** Box plot showing peak nuclear speeds in control and *Patronin* embryos; *n* = 546 and 291 in control and *Patronin*, respectively, from k = 3 embryos for each background. Scale bar = 2 μm for (B), and 5 μm for (F and G). All scatter plots show the mean ± SD. Statistical significance was calculated using the Mann–Whitney U-test. ***P < 0.001 and ****P < 0.0001.

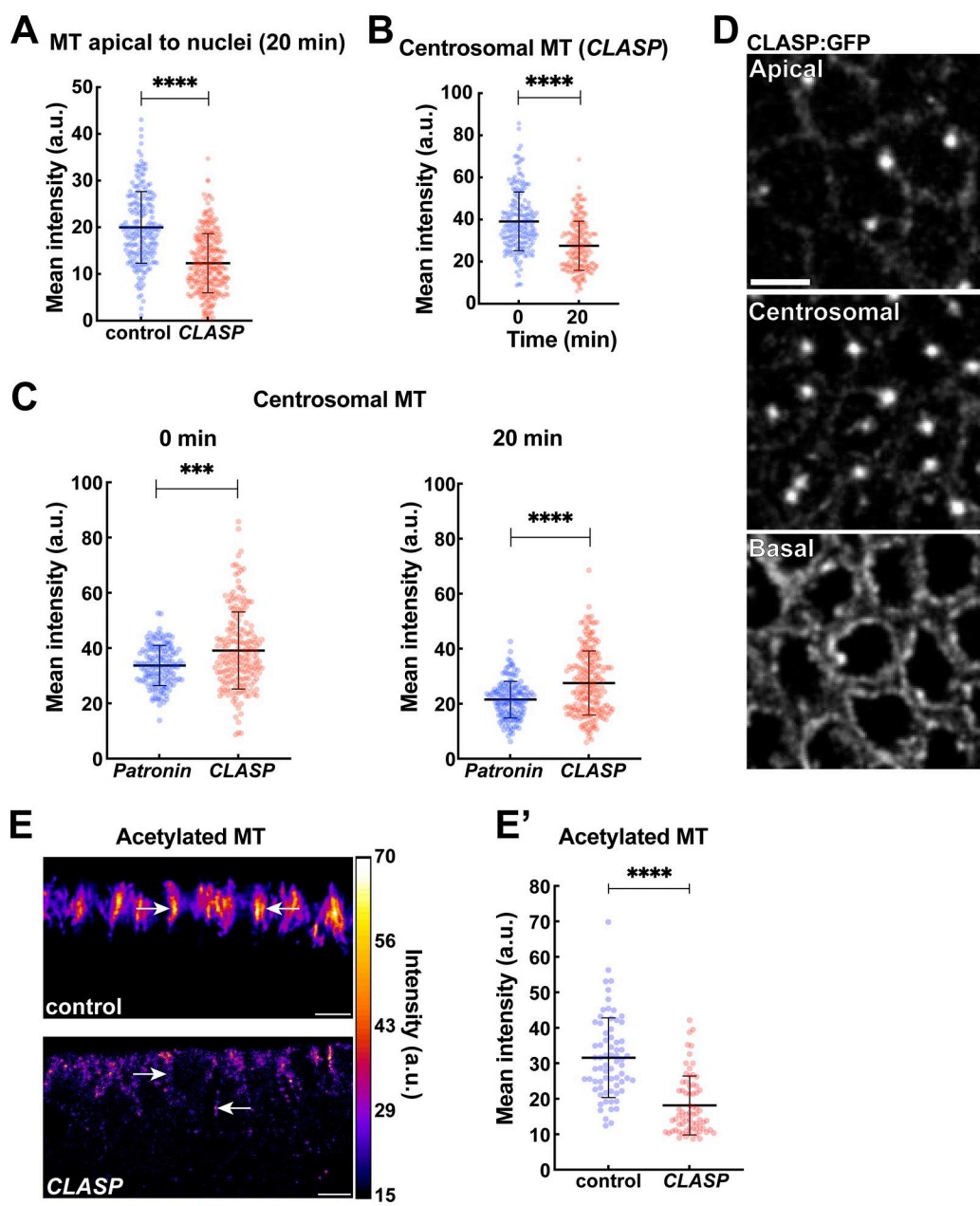

Figure S3. **CLASP disruption enhances centrosomal MTs while depleting perinuclear MT networks. (A)** Quantification showing depleted MT intensities in apical regions above nuclei in *CLASP*-compromised embryos as compared to the control at 20 min into GBE; n = 200 regions for control and n = 288 regions for *CLASP* from k = 3 embryos for each background. **(B)** Centrosomal MT intensities in *CLASP* embryos measured at 0 and 20 min GBE; n = 200 centrosomal regions for each time point from k = 4 embryos. **(C)** Centrosomal MT intensities measurement showing *CLASP* embryos have further enhancement of centrosomal MTs as compared to *Patronin* embryos; n = 150 and 200 centrosomal regions for *Patronin* and *CLASP*, respectively, at 0 min, and n = 149 and 200 centrosomal regions for *Patronin* and *CLASP*, respectively, at 20 min from k = 3 and 4 embryos for *Patronin* and *CLASP*, respectively. **(D)** Images showing CLASP:GFP localization in apical, centrosomal, and perinuclear regions at the onset of GBE. **(E)** Orthogonal view of acetylated MT color-coded for intensity levels in control and *CLASP* embryos, highlighting depleted and fragmented MT pools in *CLASP* embryos. Arrows mark acetylated MT in perinuclear regions. **(E')** Quantification of acetylated MT intensities for control and CLASP embryos (mean ± SD); n = 70 regions from k = 7 embryos for each background. Scale bar = 5 μm. All scatter plots show the mean ± SD. Statistical significance was calculated using the Mann–Whitney U-test. ***P < 0.001 and ****P < 0.0001.

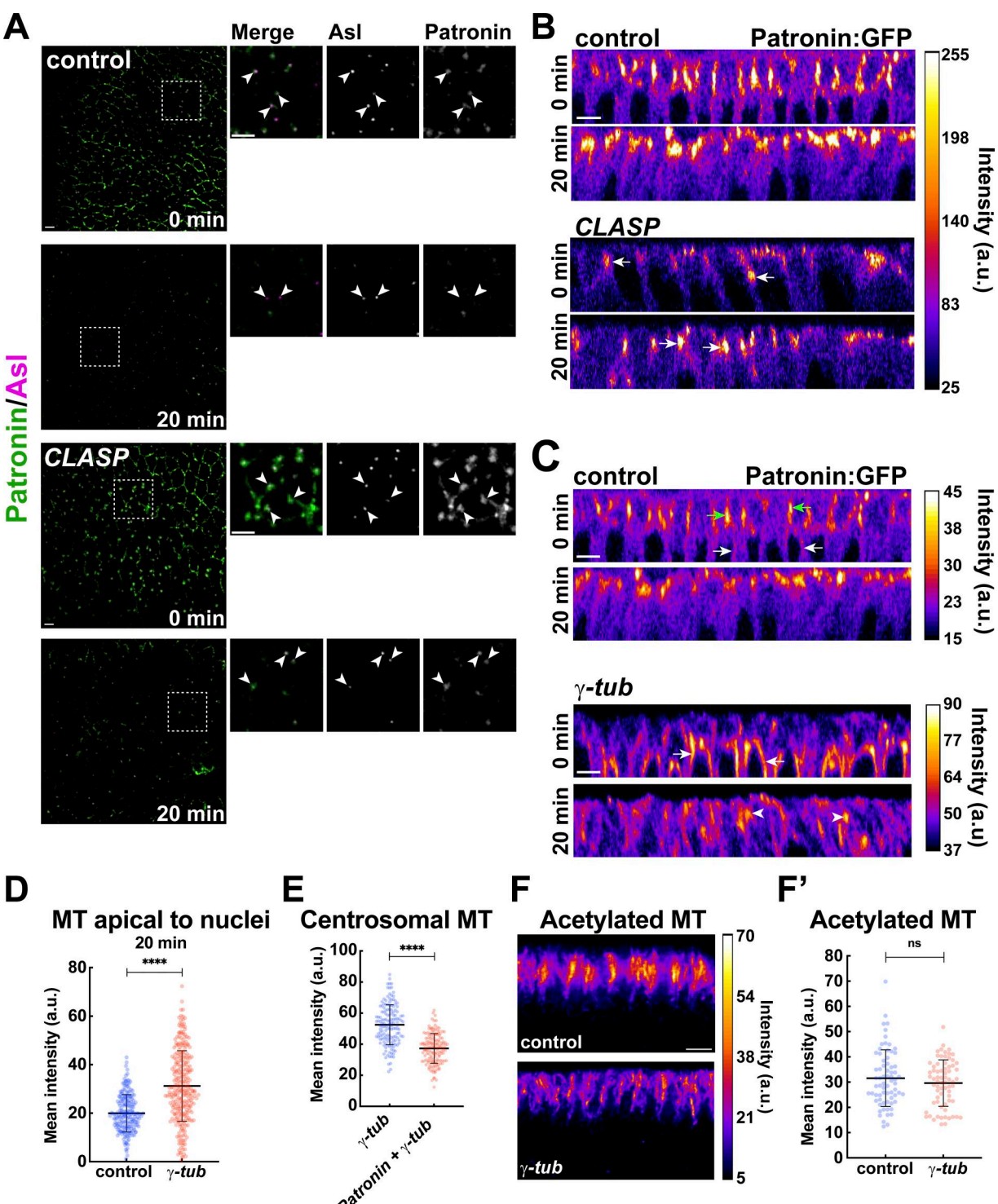

**Figure S4.** **Patronin and acetylated MT intensities in γ-tubulin37C–disrupted embryos. (A)** Still images showing Patronin:GFP enrichment at centrosomes (Asl:mCh) in control and *CLASP* embryos. Arrowheads point to the centrosomes. **(B)** Intensity heatmap for Patronin:GFP shown in orthogonal view in control and *CLASP* embryos, highlighting centrosomal enrichment of Patronin in *CLASP* embryos (arrows). **(C)** Orthogonal projection of Patronin:GFP intensity heatmap showing enhanced perinuclear (white arrows) and centrosomal (arrowheads) Patronin pools in *γ-tub* embryos as compared to control embryos. Green arrows indicate cortical Patronin. **(D)** Quantification showing MT intensities in apical regions above nuclei in *γ-tubulin*-compromised embryos as compared to the control at 20 min into GBE; *n* = 200 regions for control and *n* = 284 regions for *γ-tub* from *k* = 3 embryos for each background. **(E)** Scatter plot comparing centrosomal MT intensities in *γ-tubulin* and Patronin-*γ-tubulin* co-depleted embryos; *n* = 150 centrosomal regions from *k* = 3 embryos for each background. **(F)** Orthogonal view of acetylated MT color-coded for the intensity level in control and *γ-tub* embryos, and quantification is shown in **(F′)**; *n* = 70 regions from *k* = 7 embryos for each background. Scale bar = 5 µm. All scatter plots show the mean ± SD. Statistical significance was calculated using the Mann–Whitney U-test. ns, not significant.

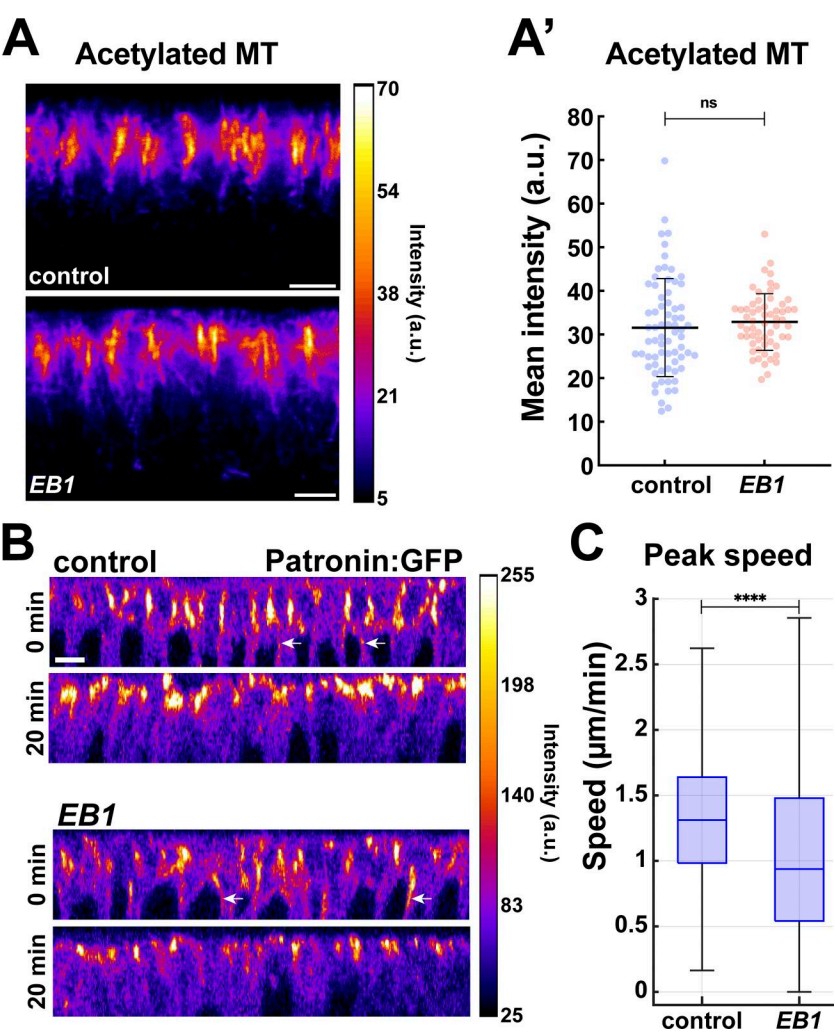

Figure S5. **Patronin and acetylated MT intensities in *EB1*-disrupted embryos. (A)** Orthogonal view of acetylated MTs color coded for intensity levels in control and *EB1* embryos, and quantification is shown in **(A')** (mean ± SD); *n* = 70 and 60 regions for control and *EB1*, respectively, from k = 7 and 6 embryos for control and *EB1*, respectively. **(B)** Orthogonal projection showing Patronin intensities at perinuclear regions (arrows) in control and *EB1* embryos. **(C)** Peak nuclear speeds in control and *EB1* embryos; *n* = 546 and 383 nuclei for control and *EB1*, respectively. Scale bar = 5 μm. Statistical significance was calculated using the Mann–Whitney U-test. ns, not significant. ****P < 0.0001.

Video 1.  **MTs undergo rapid remodeling during GBE.** Maximum-intensity projection movie of Jupiter:GFP-labeled MTs in control embryos. Centrosomal MTs are present at the beginning of the movie before undergoing detachment from the centrosomes and adopting a more cortical localization as GBE advances. Movies were acquired at 15 s per frame and displayed at 15 frames per second. Scale bar = 10 μm.

Video 2.  ***Patronin* is required to build a robust perinuclear MT network.** Maximum-intensity projection movie of Jupiter:GFP-labeled MT in control and *Patronin* shRNA embryos. A loss of ncMTs is observed while centrosomal MTs are persistent and enriched *Patronin* function is compromised. Movies were acquired at 15 s per frame and displayed at 15 frames per second. Scale bar = 10 μm.

Video 3.  ***CLASP* perturbation results in centrosomally enriched MT networks.** Maximum-intensity projection movie of Jupiter: GFP-labeled MT in control and *CLASP* shRNA embryos. A loss of ncMTs is observed while centrosomal MTs are highly enriched in *CLASP* embryos. Movies were acquired at 15 s per frame and displayed at 15 frames per second. Scale bar = 10 μm.

