## [Peer Review File · The Journal of Cell Biology]

Perinuclear non-centrosomal microtubules direct nuclei dispersion during epithelial morphogenesis

Rashmi Budhathoki, Liam Russell, Dinah Loerke, and James Blankenship

Corresponding Author(s): James Blankenship, University of Denver

Review Timeline:

Submission Date:	2025-07-16
Editorial Decision:	2025-08-16
Revision Received:	2025-09-17
Editorial Decision:	2025-09-24
Revision Received:	2025-09-30

Monitoring Editor: Mark Peifer

Scientific Editor: Dan Simon

Transaction Report:

DOI: <https://doi.org/10.1083/jcb.202507117>

August 16, 2025

Re: JCB manuscript #202507117

James Blankenship
University of Denver

Dear Todd,

Thank you for submitting your manuscript entitled "A transition to perinuclear non-centrosomal MT networks directs dispersion of nuclei during epithelial morphogenesis." The manuscript was assessed by two expert reviewers, whose comments are appended to this letter. We invite you to submit a revision if you can address the reviewers' key concerns, as outlined here.

You will see that both reviewers prize the importance of the problem and the thorough quantitative approach. However, both have some concerns that we feel can be addressed with some reasonable revisions. All comments should be addressed in some way. Reviewer 1 requests new data to explore the effects of EB1 depletion on myosin localization. Reviewer 2 has several concerns. Some simply involve a broader description of the previous literature. They suggest a need to provide stronger evidence that Patronin compensates for loss of γ -tubulin. They also request validation of RNAi constructs--in this case you should either cite previous literature validating the constructs or perform independent validation.

GENERAL GUIDELINES:

Text limits: Character count for an Article is < 40,000, not including spaces. Count includes title page, abstract, introduction, results, discussion, and acknowledgments. Count does not include materials and methods, figure legends, references, tables, or supplemental legends.

Figures: Articles may have up to 10 main text figures. Figures must be prepared according to the policies outlined in our Instructions to Authors, under Data Presentation, <https://jcb.rupress.org/site/misc/ifora.xhtml>. All figures in accepted manuscripts will be screened prior to publication.

Supplemental information: There are strict limits on the allowable amount of supplemental data. Articles may have up to 5 supplemental figures. Up to 10 supplemental videos or flash animations are allowed. A summary of all supplemental material should appear at the end of the Materials and methods section.

Please note that JCB now requires authors to submit Source Data used to generate figures containing gels and Western blots with all revised manuscripts. This Source Data consists of fully uncropped and unprocessed images for each gel/blot displayed in the main and supplemental figures. For assays performed using capillary electrophoresis and/or immunoassay-based detection, authors should instead provide the electropherogram graph(s) for each experiment, plotting fluorescence/chemiluminescence intensity vs. molecular weight/size. Please be sure to provide one Source Data file for each figure gels, blots, and/or capillary electrophoresis assays along with your revised manuscript files. File names for Source Data figures should be alphanumeric without any spaces or special characters (i.e., SourceDataF#, where F# refers to the associated main figure number or SourceDataFS# for those associated with Supplementary figures). For traditional gels and blots, the lanes of the gels/blots should be labeled as they are in the associated figure, the place where cropping was applied should be marked (with a box), and molecular weight/size standards should be labeled wherever possible. For capillary electrophoresis assays, each trace in the graph should be color-coded and labeled to indicate which protein, gene, or sample is being measured (please try to avoid red/green combinations to accommodate our color-blind readers).

The typical timeframe for revisions is three to four months. If you anticipate any difficulties in meeting this aforementioned revision time limit, please contact us and we can work with you to find an appropriate time frame for resubmission. Please note that papers are generally considered through only one revision cycle, so any revised manuscript will likely be either accepted or rejected.

Thank you for this interesting contribution to Journal of Cell Biology. You can contact us at the journal office with any questions at cellbio@rockefeller.edu.

Sincerely,

Mark Peifer, PhD
Monitoring Editor
Journal of Cell Biology

Dan Simon, PhD
Scientific Editor
Journal of Cell Biology

Reviewer #1 (Comments to the Authors (Required)):

The manuscript by Budhathoki and colleagues investigates the role of microtubules (MT) in positioning nuclei during *Drosophila* germband extension. First, the authors show that the MT cytoskeleton is reorganized over development, with net MT network distribution changing from a centriolar and perinuclear to apical, non-centrosomal above the sinking nuclei. The authors then show that Patronin and CLASP are required for perinuclear and apical MT network stability and show that Patronin is required for nuclear movement and for apical nuclear exclusion. They show a complex interdependence between Patronin, CLASP, and gamma-tubulin, with: 1) Patronin loss leading to elevated centrosomal gamma-tubulin, 2) CLASP loss leading to elevated centrosomal Patronin, and 3) gamma-tubulin loss leading to elevated centrosomal, apical, and perinuclear Patronin; suggesting antagonism between centrosomal and non-centrosomal MTs in the *Drosophila* germband.

Overall, this study provides a detailed overview of how different proteins assemble a developmentally changing MT cytoskeleton and its role in nuclear positioning, and the quality of the data and quantification were appropriate for JCB. I have some suggestions for improvements that I leave to the authors' discretion to address.

Main points:

- 1) The authors use language that imply the movement of MTs within the cell. Line 118, detach from centrosomes, Line 120, shift towards, Line 122, 163, move upwards. However, I don't see the evidence for movement over depolymerization/repolymerization shifting a limiting pool of tubulin subunits. This would be analogous to affecting different actin nucleation/stabilization pathways changing the distribution of distinct networks (PMID: 24560576). I suggest the authors modify their language with regards to movement and/or clearly state alternatives.
- 2) Did the authors consider that their MT perturbations affect the actin cytoskeleton or junctions? In particular EB1 depletion could affect RhoGEF2 localization and contractility (PMID: 15498490, 36440630), which has been shown to affect nuclear positioning in other tissues (PMID: 23134725). I would suggest showing whether the EB1 depletion affects myosin localization, which would report on where Rho1 is getting activated and RhoGEF2 recruited.

Minor points:

- 1) Line 129-130, Are apical MTs acetylated?
- 2) Line 147, Fig. 2A: It is unclear from the images how 'detached' MT bundles look different from regular MT bundles.
- 3) The apical MT population changes are not quantified in Fig. 2C, 3C, 4, and 5. Because this is an important result and is quantified for EB1, I suggest including quantification.
- 4) Line 159-160: Patronin redistribution as tissue extension precedes - it would be helpful to know the precise timing of Patronin redistribution with respect to phases of extension and/or other morphological features of the embryo.
- 5) Lines 283-285: Are not centrosomal MT networks also present in gamma-tubulin knock-down (Fig. 5F)? In which case any of these populations could mediate nuclear positioning.

Reviewer #2 (Comments to the Authors (Required)):

In their submission, Budhathoki and co-workers examine microtubule organization during *Drosophila* embryo germband extension (GBE), a developmental stage leading to cell intercalation. Prior to GBE, the cells are arranged in an epithelial sheet with the microtubules organized as an inverted "basket." These microtubules are nucleated from the pair of centrosomes situated on top of the columnar nuclei. As GBE commences, the microtubules reorganize, such that more microtubules are nucleated from acentrosomal pathways. This microtubule reorganization facilitates displacement of the nuclei away from the apical cortex, which is permissive for GBE. The authors investigate how microtubules are remodeled from the centrosome-to-non-centrosomal organization through depletion of several microtubule-binding proteins, including Patronin, CLASP, EB1, and gTub 37C. Consistent with work in other systems, the authors observe levels of antagonism and compensation from different microtubule nucleation pathways. Strengths of this study include ample quantitative image analysis, examination of an understudied developmental process, and investigation of cell state transitions. Some weaknesses are noted, however. For all the depletion studies, a single RNAi construct was examined, and it is unclear if those lines are validated. Moreover, compensation and redundancy in microtubule nucleation has been previously reported, raising questions if the work is of a significantly broad conceptual advance to warrant consideration in JCB.

Major points

1. The authors fail to acknowledge prior work examined MT-nucleation pathways and their redundancy in mitotic embryos (<https://pubmed.ncbi.nlm.nih.gov/24389063/>) and in neural stem cells (<https://pmc.ncbi.nlm.nih.gov/articles/PMC10497398/>). The idea that microtubule organization shifts from centrosomes to acentrosomal pathways has also been examined previously.
2. Prior work by Tano Gonzalez's lab is also not noted here. gTub 37C was mapped and characterized in this work: <https://pubmed.ncbi.nlm.nih.gov/9155007/>. In that paper, the authors show that eggs laid by mutant mothers have severely defective early mitotic divisions and nucleate "aster-less" microtubules. These findings lead one to surmise the gTub 37C RNAi used in this work is likely an incomplete knockdown.
3. The authors suggest in the text and discussion that Patronin compensates for loss of gTub 37C based on their observation that Patronin fluorescence enriches at centrosomes upon gTub 37C RNAi. This conclusion is not supported by the available data. The localization of a protein does not establish its function. The localization of other proteins may also be altered. While it is fine for the authors to speculate in the discussion, if they wish to claim their "data indicate" their model is true (lines 282-3), they need to test it through a functional assay; for example, by co-depleting gTub 37C and Patronin. Based on the published work by the Gonzalez lab, it is conceivable that gTub 23C or indeed another factor (e.g., TPX2, MSPS, and others) may be responsible for nucleating microtubules at centrosomes partially depleted of gTub 37C.
4. The authors examine Patronin, CLASP, gTub, and Eb1 function through RNAi, and only a single transformant is used. The authors must list the exact transformant used in their Methods, presently this is only done for Patronin. A Bloomington identifier is insufficient. They must cite or show evidence that the RNAi does actually deplete the RNA or protein of interest. Finally, a single RNAi alone does not provide particularly compelling genetic evidence of a gene's function. This approach reduces the rigor and reproducibility of the conclusions.

Minor Points

1. The authors describe centrosomal localization of Patronin and CLASP, but through their figures, it is difficult to deduce how they determined where the centrosomes are. It is only upon reading the methods that we learn they used an ASL counter label. The authors should add the merge images to the appropriate figures so that the reader can see which foci of fluorescence are centrosomes or not. An example is Fig 4E, where multiple arrows point to a few concentrations of fluorescence, but other large foci are not marked. In the same figure, it is unclear where are the centrosomes in the control image.
2. That loss of CLASP leads to more Patronin at the centrosome suggests CLASP may function to displace Patronin. Do the authors see less Patronin at the centrosome when they express their GFP-CLASP transgene?
3. Figure 4 legend; several panels are described in the legend but absent from the figure (I-M).
4. The methods describe use of several drugs (lines 409-413), but those are not presented in this study.

Reviewer #1 (Comments to the Authors (Required)):

The manuscript by Budhathoki and colleagues investigates the role of microtubules (MT) in positioning nuclei during *Drosophila* germband extension. First, the authors show that the MT cytoskeleton is reorganized over development, with net MT network distribution changing from a centriolar and perinuclear to apical, non-centrosomal above the sinking nuclei. The authors then show that Patronin and CLASP are required for perinuclear and apical MT network stability and show that Patronin is required for nuclear movement and for apical nuclear exclusion. They show a complex interdependence between Patronin, CLASP, and gamma-tubulin, with: 1) Patronin loss leading to elevated centrosomal gamma-tubulin, 2) CLASP loss leading to elevated centrosomal Patronin, and 3) gamma-tubulin loss leading to elevated centrosomal, apical, and perinuclear Patronin; suggesting antagonism between centrosomal and non-centrosomal MTs in the *Drosophila* germband.

Overall, this study provides a detailed overview of how different proteins assemble a developmentally changing MT cytoskeleton and its role in nuclear positioning, and the quality of the data and quantification were appropriate for JCB. I have some suggestions for improvements that I leave to the authors' discretion to address.

Main points:

1) The authors use language that imply the movement of MTs within the cell. Line 118, detach from centrosomes, Line 120, shift towards, Line 122, 163, move upwards. However, I don't see the evidence for movement over depolymerization/repolymerization shifting a limiting pool of tubulin subunits. This would be analogous to affecting different actin nucleation/stabilization pathways changing the distribution of distinct networks (PMID: 24560576). I suggest the authors modify their language with regards to movement and/or clearly state alternatives.

Thank you for the comments and critiques. And, yes, this is a fair point – we have edited the manuscript as suggested.

2) Did the authors consider that their MT perturbations affect the actin cytoskeleton or junctions? In particular EB1 depletion could affect RhoGEF2 localization and contractility (PMID: 15498490, 36440630), which has been shown to affect nuclear positioning in other tissues (PMID: 23134725). I would suggest showing whether the EB1 depletion affects myosin localization, which would report on where Rho1 is getting activated and RhoGEF2 recruited.

Yes, we were interested in these data as well, but concentrated on the impact of EB1 disruption on nuclei and MT networks. We have also previously examined Rok-inhibited embryos (de Leeuw et al., 2024) which showed a paralysis of both the epithelium and nuclear movements – EB1 perturbed embryos (at least at this level of shRNA disruption) were much more mobile and did not possess the defects that Rok disrupted embryos displayed.

Minor points:

1) Line 129-130, Are apical MTs acetylated?

MTs at the apical cortex are not acetylated, whereas MTs positioned apical to the nuclei do show acetylation. We have included a new figure panel demonstrating this (Supplemental Fig. 1D').

2) Line 147, Fig. 2A: It is unclear from the images how 'detached' MT bundles look different from regular MT bundles.

Yes, thanks for bringing this to our attention – we agree that the example was not very representative of the phenotype (we had decreased the nuclei intensities to make it easier to see MTs, but this had the effect of eroding the apparent nuclear boundary). We have leveled the nuclei to best show nuclear dimensions in the updated figures, and the detachment phenotype should be more visible.

3) The apical MT population changes are not quantified in Fig. 2C, 3C, 4, and 5. Because this is an important result and is quantified for EB1, I suggest including quantification.

We appreciate this comment – as suggested, we have added quantifications of the apical MT populations for each background (see Supplemental Fig. 2A, 3A, and 4D).

4) Line 159-160: Patronin redistribution as tissue extension precedes - it would be helpful to know the precise timing of Patronin redistribution with respect to phases of extension and/or other morphological features of the embryo.

Yes, fair point – Patronin redistribution in the mesoderm and ectoderm was characterized in Ko et al., 2019 (although this was in relation to ventral furrow formation), which is why we did not further characterize these changes. However, we have added measurements of cortical Patronin:GFP intensities at 0 and 20 min into germband extension. These data align with the previous observation on Patronin (Supplemental Fig. 2G').

5) Lines 283-285: Are not centrosomal MT networks also present in gamma-tubulin knock-down (Fig. 5F)? In which case any of these populations could mediate nuclear positioning.

Yes, this is correct – as the main point of this section was to examine the potential antagonism between ncMT and centrosomal MTs, we have edited the summary sentences of this paragraph to reflect on this.

Reviewer #2 (Comments to the Authors (Required)):

In their submission, Budhathoki and co-workers examine microtubule organization during *Drosophila* embryo germband extension (GBE), a developmental stage leading to cell intercalation. Prior to GBE, the cells are arranged in an epithelial sheet with the microtubules organized as an inverted "basket." These microtubules are nucleated from the pair of centrosomes situated on top of the columnar nuclei. As GBE commences, the microtubules reorganize, such that more microtubules are nucleated from acentrosomal pathways. This microtubule reorganization facilitates displacement of the nuclei away from the apical cortex, which is permissive for GBE. The authors investigate how microtubules are remodeled from the centrosome-to-non-centrosomal organization through depletion of several microtubule-binding proteins, including Patronin, CLASP, EB1, and gTub 37C. Consistent with work in other systems, the authors observe levels of antagonism and compensation from different microtubule nucleation pathways. Strengths of this study include ample quantitative image analysis, examination of an understudied developmental process, and investigation of cell state transitions. Some weaknesses are noted, however. For all the depletion studies, a single RNAi construct was examined, and it is unclear if those lines are validated. Moreover, compensation and redundancy in microtubule nucleation has been previously reported, raising questions if the work is of a significantly broad conceptual advance to warrant consideration in JCB.

Major points:

1. The authors fail to acknowledge prior work examined MT-nucleation pathways and their redundancy in mitotic embryos (<https://pubmed.ncbi.nlm.nih.gov/24389063/>) and in neural stem cells (<https://pmc.ncbi.nlm.nih.gov/articles/PMC10497398/>). The idea that microtubule organization shifts from centrosomes to acentrosomal pathways has also been examined previously.

Thanks for pointing out these works. We have now incorporated the references into the discussion section.

2. Prior work by Tano Gonzalez's lab is also not noted here. gTub 37C was mapped and characterized in this work: <https://pubmed.ncbi.nlm.nih.gov/9155007/>. In that paper, the authors show that eggs laid by mutant mothers have severely defective early mitotic divisions and nucleate "aster-less" microtubules. These findings lead one to surmise the gTub 37C RNAi used in this work is likely an incomplete knockdown.

Yes, we agree that the γ -tub37C RNAi used in this work is likely an incomplete knockdown. This was intentional, as a complete loss of γ -tub37C function causes severe early mitotic defects and prevents development beyond the earliest embryonic stages. By using partial knockdown, we were able to allow embryos to progress to gastrulation, which was essential for addressing the specific

questions in this study. We wanted to examine if compensation was apparent when gamma-tubulin function was disrupted, so a partial knock down was sufficient (it showed a strong effect on the studied MT populations) and a full loss-of-function was not required.

3. The authors suggest in the text and discussion that Patronin compensates for loss of gTub 37C based on their observation that Patronin fluorescence enriches at centrosomes upon gTub 37C RNAi. This conclusion is not supported by the available data. The localization of a protein does not establish its function. The localization of other proteins may also be altered. While it is fine for the authors to speculate in the discussion, if they wish to claim their "data indicate" their model is true (lines 282-3), they need to test it through a functional assay; for example, by co-depleting gTub 37C and Patronin. Based on the published work by the Gonzalez lab, it is conceivable that gTub 23C or indeed another factor (e.g., TPX2, MSPS, and others) may be responsible for nucleating microtubules at centrosomes partially depleted of gTub 37C.

A fair point – we agree that the double depletion was needed to state the above conclusively. To this end, in new data we examined centrosomal MT intensities in embryos with γ -tub37C and Patronin co-depleted. In this new analysis, MT intensities decrease in co-depleted embryos compared to single γ -tub37C-depleted embryos. We have added these results (Supplemental Fig. 4E) and edited the discussion on this point.

4. The authors examine Patronin, CLASP, gTub, and Ebl function through RNAi, and only a single transformant is used. The authors must list the exact transformant used in their Methods, presently this is only done for Patronin. A Bloomington identifier is insufficient. They must cite or show evidence that the RNAi does actually deplete the RNA or protein of interest. Finally, a single RNAi alone does not provide particularly compelling genetic evidence of a gene's function. This approach reduces the rigor and reproducibility of the conclusions.

These shRNA lines have been validated and used in previous studies – we have added references and transgene identifiers for each of them in the Methods section.

Minor Points:

1. The authors describe centrosomal localization of Patronin and CLASP, but through their figures, it is difficult to deduce how they determined where the centrosomes are. It is only upon reading the methods that we learn they used an ASL counter label. The authors should add the merge images to the appropriate figures so that the reader can see which foci of fluorescence are centrosomes or not. An example is Fig 4E, where multiple arrows point to a few concentrations of fluorescence, but other large foci are not marked. In the same figure, it is unclear where are the centrosomes in the control image.

We have now added a figure panel with Asl:mCh marking centrosomes in both control and CLASP embryos (See Supplemental Fig. 4A).

2. That loss of CLASP leads to more Patronin at the centrosome suggests CLASP may function to displace Patronin. Do the authors see less Patronin at the centrosome when they express their GFP-CLASP transgene?

An interesting experiment – however, we were unable to test this overexpression experiment due to an incompatibility with our required markers.

3. Figure 4 legend; several panels are described in the legend but absent from the figure (I-M).

We are sorry, but we carefully checked the Figure 4 legend and did not find a description of panels that were absent from the figure.

4. The methods describe the use of several drugs (lines 409-413), but those are not presented in this study.

Thank you for catching this – we have edited the Methods.

September 24, 2025

RE: JCB Manuscript #202507117R

James Blankenship
University of Denver

Dear Prof. Blankenship,

Thank you for submitting your revised manuscript entitled "Perinuclear non-centrosomal MT networks direct dispersion of nuclei during epithelial morphogenesis." We would be happy to publish your paper in JCB pending final revisions necessary to meet our formatting guidelines (see details below).

A. MANUSCRIPT ORGANIZATION AND FORMATTING:

1) Text limits: Character count for Articles is < 40,000, not including spaces. Count includes title page, abstract, introduction, results, discussion, and acknowledgments. Count does not include materials and methods, figure legends, references, tables, or supplemental legends.

2) Figure formatting: Articles may have up to 10 main text figures. Scale bars must be present on all microscopy images, including inset magnifications. Also, please avoid pairing red and green for images and graphs to ensure legibility for color-blind readers. If red and green are paired for images, please ensure that the particular red and green hues used in micrographs are distinctive with any of the colorblind types. If not, please modify colors accordingly or provide separate images of the individual channels.

3) Statistical analysis: Error bars on graphic representations of numerical data must be clearly described in the figure legend. The number of independent data points (n) represented in a graph must be indicated in the legend. Please indicate whether 'n' refers to technical or biological replicates (i.e. number of analyzed cells, samples or animals, number of independent experiments). If independent experiments with multiple biological replicates have been performed, we recommend using distribution-reproducibility SuperPlots (please see Lord et al., JCB 2020) to better display the distribution of the entire dataset, and report statistics (such as means, error bars, and P values) that address the reproducibility of the findings.

Statistical methods should be explained in full in the materials and methods. For figures presenting pooled data the statistical measure should be defined in the figure legends. Please also be sure to indicate the statistical tests used in each of your experiments (both in the figure legend itself and in a separate methods section) as well as the parameters of the test (for example, if you ran a t-test, please indicate if it was one- or two-sided, etc.). Also, if you used parametric tests, please indicate if the data distribution was tested for normality (and if so, how). If not, you must state something to the effect that "Data distribution was assumed to be normal but this was not formally tested."

4) Title: The title should be less than 100 characters including spaces and should not use abbreviations such as "MT." We suggest revising the title to: "Perinuclear non-centrosomal microtubules direct nuclei dispersion during epithelial morphogenesis" or something similar.

5) Materials and methods: Should be comprehensive and not simply reference a previous publication for details on how an experiment was performed. Please provide full descriptions (at least in brief) in the text for readers who may not have access to referenced manuscripts. The text should not refer to methods "...as previously described."

6) For all cell lines, vectors, strains, constructs/cDNAs, etc. - all genetic material: please include database / vendor ID (e.g. Addgene, ATCC, etc.) or if unavailable, please briefly describe their basic genetic features, even if described in other published work or gifted to you by other investigators (and provide references where appropriate). Please be sure to provide the sequences for all of your oligos: primers, si/shRNA, RNAi, gRNAs, etc. in the materials and methods. You must also indicate in the methods the source, species, and catalog numbers/vendor identifiers (where appropriate) for all of your antibodies, including secondary. If antibodies are not commercial, please add a reference citation if possible.

7) Microscope image acquisition: The following information must be provided about the acquisition and processing of images:
a. Make and model of microscope

- b. Type, magnification, and numerical aperture of the objective lenses
- c. Temperature
- d. Imaging medium
- e. Fluorochromes
- f. Camera make and model
- g. Acquisition software
- h. Any software used for image processing subsequent to data acquisition. Please include details and types of operations involved (e.g., type of deconvolution, 3D reconstitutions, surface or volume rendering, gamma adjustments, etc.).

8) References: There is no limit to the number of references cited in a manuscript. References should be cited parenthetically in the text by author and year of publication. Abbreviate the names of journals according to PubMed.

9) Supplemental materials: Articles may have up to 5 supplemental figures and 10 videos. Please also note that tables, like figures, should be provided as individual, editable files. A summary of all supplemental material should appear at the end of the Materials and methods section. Please include one brief sentence per item.

10) Video legends: Should describe what is being shown, the cell type or tissue being viewed (including relevant cell treatments, concentration and duration, or transfection), the imaging method (e.g., time-lapse epifluorescence microscopy), what each color represents, how often frames were collected, the frames/second display rate, and the number of any figure that has related video stills or images.

11) eTOC summary: A ~40-50 word summary that describes the context and significance of the findings for a general readership should be included on the title page. The statement should be written in the present tense and refer to the work in the third person. It should begin with "First author name(s) et al..." to match our preferred style.

13) A separate author contribution section is required following the Acknowledgments in all research manuscripts. All authors should be mentioned and designated by their first and middle initials and full surnames. We encourage use of the CRediT nomenclature (<https://casrai.org/credit/>).

14) ORCID IDs: ORCID IDs are unique identifiers allowing researchers to create a record of their various scholarly contributions in a single place. Please note that ORCID IDs are required for all authors. At resubmission of your final files, please be sure to provide your ORCID ID and those of all co-authors.

15) Journal of Cell Biology now requires a data availability statement for all research article submissions. These statements will be published in the article directly above the Acknowledgments. The statement should address all data underlying the research presented in the manuscript. Please visit the JCB instructions for authors for guidelines and examples of statements at (<https://rupress.org/jcb/pages/editorial-policies#data-availability-statement>).

B. FINAL FILES:

Thank you for your attention to these final processing requirements. Please revise and format the manuscript and upload materials within 7 days. If you need an extension for whatever reason, please let us know and we can work with you to determine a suitable revision period.

Thank you for this interesting contribution, we look forward to publishing your paper in Journal of Cell Biology.

Sincerely,

Mark Peifer, PhD
Monitoring Editor
Journal of Cell Biology

Dan Simon, PhD
Scientific Editor
Journal of Cell Biology